# Prolonged partner separation erodes nucleus accumbens transcriptional signatures of pair bonding in male prairie voles

Julie M Sadino[1], Xander G Bradeen[2,3], Conor J Kelly[1,4], Liza E Brusman[1], Deena M Walker[5], Zoe R Donaldson[1,2]*

[1]Department of Molecular, Cellular, and Developmental Biology, University of Colorado Boulder, Boulder, United States; [2]Department of Psychology and Neuroscience, University of Colorado Boulder, Boulder, United States; [3]Department of Adult Hematology, University of Colorado- Anschutz Medical Campus, Aurora, United States; [4]BioFrontiers Institute, University of Colorado Boulder, Boulder, United States; [5]Department of Behavioral Neuroscience, Oregon Health and Science University, School of Medicine, Portland, United States

*For correspondence: zoe.donaldson@colorado.edu

**Abstract** The loss of a spouse is often cited as the most traumatic event in a person's life. However, for most people, the severity of grief and its maladaptive effects subside over time via an understudied adaptive process. Like humans, socially monogamous prairie voles (*Microtus ochrogaster*) form opposite-sex pair bonds, and upon partner separation, show stress phenotypes that diminish over time. We test the hypothesis that extended partner separation diminishes pair bond-associated behaviors and causes pair bond transcriptional signatures to erode. Opposite-sex or same-sex paired males were cohoused for 2 weeks and then either remained paired or were separated for 48 hours or 4 weeks before collecting fresh nucleus accumbens tissue for RNAseq. In a separate cohort, we assessed partner-directed affiliation at these time points. We found that these behaviors persist despite prolonged separation in both same-sex and opposite-sex paired voles. Opposite-sex pair bonding led to changes in accumbal transcription that were stably maintained while animals remained paired but eroded following prolonged partner separation. Eroded genes are associated with gliogenesis and myelination, suggesting a previously undescribed role for glia in pair bonding and loss. Further, we pioneered neuron-specific translating ribosomal affinity purification in voles. Neuronally enriched transcriptional changes revealed dopaminergic-, mitochondrial-, and steroid hormone signaling-associated gene clusters sensitive to acute pair bond disruption and loss adaptation. Our results suggest that partner separation erodes transcriptomic signatures of pair bonding despite core behavioral features of the bond remaining intact, revealing potential molecular processes priming a vole to be able to form a new bond.

## Editor's evaluation

This study focuses on an important but understudied topic, the biological basis of pair bonding and associated stress when the bond is lost. The authors present convincing evidence that transcriptional changes associated with the formation of a pair bond in voles degrade during prolonged separation but do not precisely mirror behavioral responses. Gene expression changes were associated with glia, steroid hormone signaling, and myelination, among others, providing critical guideposts for

future investigations. This work will be of interest to behavioral neuroscientists and/or those interested in social behavior.

## Introduction

The death of a romantic partner is cited as one of the most traumatic experiences in a person's life and results in grief and corresponding indicators of mental and physiological distress (*Keyes et al., 2014*; *Holmes and Rahe, 1967*). However, for the majority of people acute grief subsides within approximately six months as the bereaved integrates and adapts to the loss (*Shear et al., 2011*). Most people will also eventually form a new pair bond, which provides a behavioral indicator of loss adaptation (*Shear and Shair, 2005*). The processes that enable such adaptation remain poorly understood but likely occur via the same neural systems that are uniquely engaged by pair bonding (*Blocker and Ophir, 2016*).

Socially monogamous prairie voles are a laboratory-amenable species that recapitulates many aspects of human social bonds, making them ideal for interrogating the neurobiology of bonding and loss. Pair bonding in this species results in a shift in sociobehavioral state where specific behaviors are only evident after the transition from living in a same-sex pair (i.e. family group) to an opposite-sex bonded pair. Pair bonded voles will prefer to affiliate with their partner, display selective aggression towards non-partner individuals, and exhibit robust and organized biparental care (*Williams, 1992*; *Getz et al., 1981*; *Williams et al., 1992*; *Insel and Young, 2001*; *Carter et al., 1995*; *Winslow et al., 1993*). Considering that other stable behavioral shifts, such as reproductive status and dominance hierarchies, are underwritten by changes in transcription, we aimed to determine if a mature pair bond—and the process of adapting to loss—are underwritten by stable changes in gene expression in the nucleus accumbens (*Cardoso et al., 2015*; *Zayed and Robinson, 2012*; *Tripp et al., 2018*).

While an extended network of brain regions mediates social processing and decision making, the nucleus accumbens (NAc)—a region important for reward, motivation, and action selection—is a critical hub that is engaged when forming a bond and is implicated in loss processing (*Aragona and Wang, 2004*; *Lim and Young, 2004*; *Walum and Young, 2018*). In humans, holding hands with a romantic partner enhanced blood oxygenation levels (BOLD) in the NAc and successful adaptation to spousal loss is associated with a reduced partner-associated BOLD signal in this region (*O'Connor et al., 2008*; *Kreuder et al., 2017*). Thus, the NAc contributes to pair bonding and loss and broad scale changes in this brain region potentially mediate shifts in transcription which may be required to adapt to partner loss. Similarly, in prairie voles, pair bond-associated behavioral changes are underwritten by several known neuromolecular changes within the NAc which maintain and reinforce pair bonds over time (*Insel and Young, 2001*; *Aragona and Wang, 2004*; *Young and Wang, 2004*). Further, when separated from their partner, prairie voles exhibit behavioral and physiological distress that mirrors what is seen upon bond disruption in humans and other species, such as titi monkeys (*Carter et al., 1995*). These include increased circulating glucocortioids levels, activation of the HPA axis, increased anxiety, decreased pain thresholds, and autonomic dysfunction (*McNeal et al., 2014*; *Grippo et al., 2007*; *Bosch et al., 2016*; *Pohl et al., 2019*; *Osako et al., 2018*). However, we have previously shown that if a pair bonded prairie vole loses their partner, given enough time, they will form a new bond that supersedes the original bond (*Harbert et al., 2020*; *Tamarin et al., 1990*). As in humans, the ability to form a new bond indicates that prairie voles can adapt to partner loss.

Here, we map the trajectory of the pair bond transcriptional profile by comparing opposite-sex paired males to same-sex sibling male pairs. In the wild, sexually naive male voles can cohabitate with other males, especially siblings, although these relationships do not persist after males form an opposite-sex pair bond and establish a nest/territory with their partner (*Getz et al., 1981*; *Carter et al., 1995*). By comparing opposite-sex pair bonded voles to their same-sex paired counterparts before and after partner separation, we have an ethologically relevant means to isolate the unique biology of bonding and loss independent of those that support general affiliative interactions (e.g. peer relationships) or the stressful effects of social isolation more broadly. Thus, we compared behavior and NAc transcriptional profiles of opposite-sex pair bonded and same-sex cohoused naïve male voles to explicitly define the pair bond both behaviorally and transcriptionally. Then we examined how affiliative pair bond characteristics were altered by partner separation to test the hypothesis that pair bond transcriptional signatures and bond-associated behaviors would erode as a function of time since

**eLife digest** Losing a spouse or life partner is a deeply traumatic event that can have long-term repercussions. Given enough time, however, most surviving partners are able to process their grief. The neural processes that enable people to adapt to their loss remain unknown.

To explore this question, scientists often turn to animals that form long-term mating based pair bonds and can be raised in the laboratory. Monogamous prairie voles enter lifelong partnerships where the two individuals live together, prefer to cuddle with each other, and take care of their pups as a team. After having lost their mate, they show signs of distress that eventually subside with time.

Sadino et al. examined the biological impact of partner loss in these animals by focusing on the nucleus accumbens, a brain region important for social connections. This involved tracking gene expression – which genes were switched on and off in this area – as the voles established their pair bonds, and then at different time points after one of the partners had been removed.

The experiments revealed that establishing a relationship leads to a stable shift in nucleus accumbens gene expression, which may help maintain bonds over time. In particular, genes related to glia (the non-neuronal cells which assist neurons in their tasks) see their expression levels increase, indicating a previously undescribed role for this cell type in regulating pair bonding. Having their partner removed led to an erosion of the gene expression pattern that had emerged during pair bonding; this may help the remaining vole adapt to its loss and go on to form a new bond. In addition, Sadino et al. explored the gene expression of only neurons in the nucleus accumbens and uncovered biological processes distinct from those that occur in glia after partner separation. Together, these results shed light on the genetic and neuronal mechanisms which underlie adaptation to loss; this knowledge could one day inform how to better support individuals during this time.

partner separation. We reasoned that these changes represent key components of loss adaptation that, together, may prime the vole to be able to form a new bond.

We found that pairing induced a reliable affiliative preference for a peer or a pair bonded partner, and that this preference is remarkably stable, persisting even after four weeks of separation. We further show that pair bond-associated changes in accumbal gene expression were consistent at two and six weeks post-pairing. However, once opposite-sex pairs are separated, the pair bond transcriptional signature erodes as a function of separation time. To further home in on the transcriptional changes associated with partner separation specifically in NAc neurons, we pioneered translating ribosomal affinity purification in voles (vTRAP). Using vTRAP, we identified clusters of genes associated with dopaminergic signaling, mitochondrial organization, and steroid hormone signaling whose expression patterns are sensitive to acute pair bond disruption and loss adaptation. In sum, our parallel behavioral and transcriptional data suggests that erosion of pair bond transcriptional signatures in the NAc precedes changes in affiliative partner preference, providing insight into time-dependent neuromolecular changes that may contribute to loss adaptation.

## Results

We determined how bonding and extended separation affects social behavior and NAc transcription in opposite-sex and same-sex paired males. We employed timepoints that are experimentally validated and ethologically relevant (*Figure 1A*). We paired all study animals for 2 weeks, a duration that reliably produces mature pair bonds (*Scribner et al., 2020*; *Brusman et al., 2021*). For the separation timepoints, the ability to form a new bond after loss serves as a behavioral metric of loss adaptation. Prior work has shown that male voles are able to form a new pair bond 4 weeks post-separation, but not 2 weeks or earlier (*Harbert et al., 2020*).

Consistent with our laboratory observations, wild male voles who lose a partner will remain at the nest for ~17 days (*Tamarin et al., 1990*). Thus, we assessed behavior and transcription in paired voles two days and four weeks after separation—timepoints before and after the animal can form a new bond.

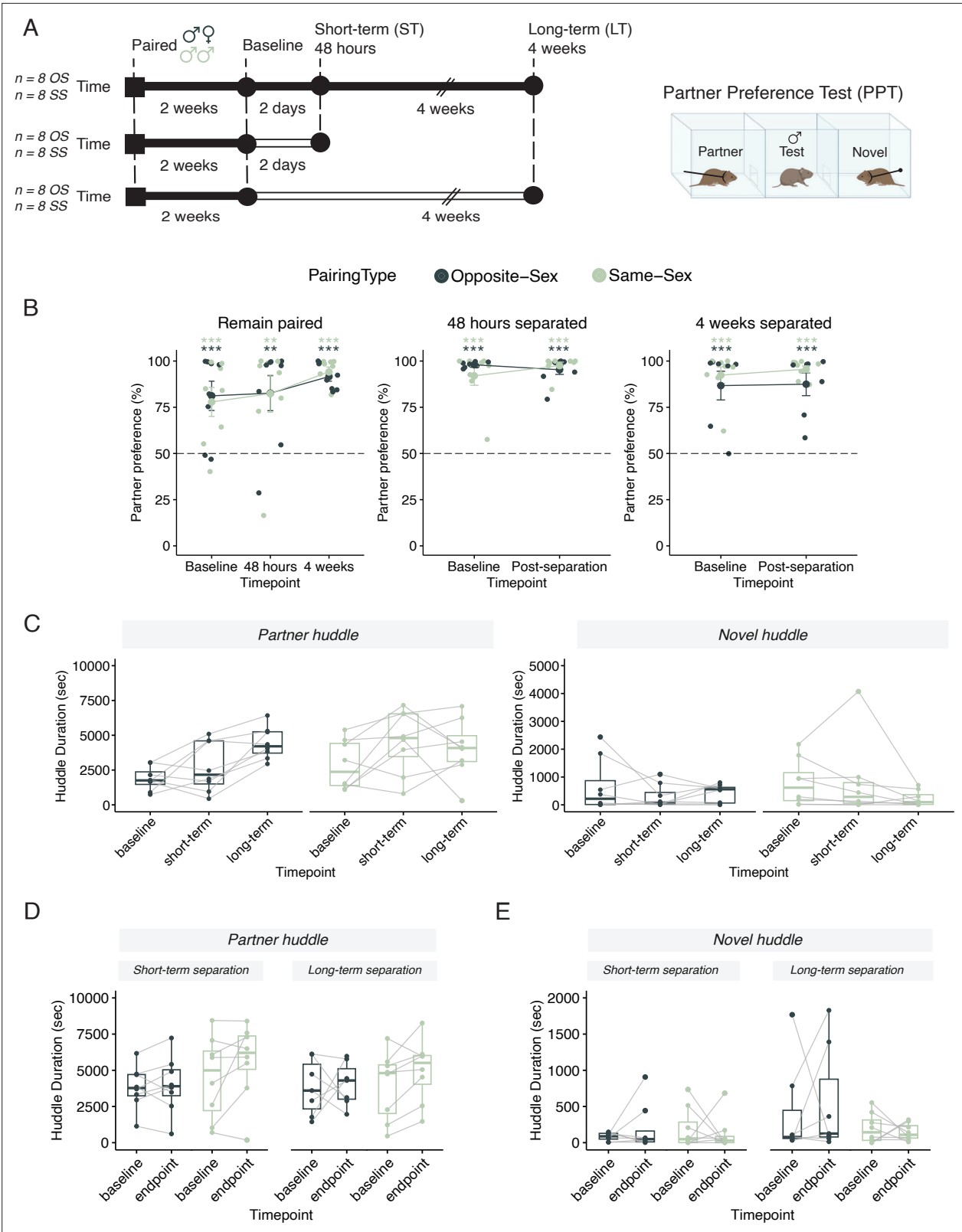

**Figure 1.** Males in either opposite-sex or same-sex pairs retain their partner preferences for at least four weeks following partner separation. Additional stats including post-hocs provided in *Supplementary file 1*. (**A**) Schematic of a partner preference test (PPT) and timelines for behavioral experiments. Opposite-sex (n=8/cohort) and same-sex (n=8/cohort) pairs were paired for 2 weeks prior to a baseline PPT. Pairs were then either in a remain paired or separated cohort. Remain paired animals stayed with their partners throughout the experiment and underwent PPT 48 hours and 4 weeks after baseline.

*Figure 1 continued on next page*

*Figure 1 continued*

Separated animals were separated from their partners for either 48 hours (short-term) or 4 weeks (long-term) prior to the endpoint PPT. (**B**) Partner preference scores (% partner huddle/total huddle) from baseline, short-term, and long-term PPTs of opposite-sex (dark green) and same-sex (light green) paired males. Opposite- and same-sex paired males showed a baseline partner preference that remained evident for both groups after short-term separation and long-term separation (one-way t-tests relative to 50%, *** p<0.001, ** p<0.01). Black dotted line at 50% indicates no preference for partner or novel. (**C**) Partner (left) and novel (right) huddle duration (seconds) for remain paired animals over time during each PPT (two-way RM-ANOVA). (**D**) Partner huddle for short-term (left) and long-term (right) separated animals between baseline and the endpoint PPT—48 hours or 4 weeks post-baseline, respectively (two-way RM-ANOVA). (**E**) Novel huddle for short-term (left) and long-term (right) separated animals between baseline and the endpoint PPT—48 hours or 4 weeks post-baseline respectively (two-way RM-ANOVA).

The online version of this article includes the following figure supplement(s) for figure 1:

**Figure supplement 1.** Partner and novel huddle latency is unchanged in both remain paired and separated opposite- and same-sex paired males.

**Figure supplement 2.** Change scores of partner preference and partner huddle duration for opposite-sex and same-sex paired males.

**Figure supplement 3.** Pearson's correlation matrices of partner preference test behaviors across time points.

## Extended separation disrupts the strengthening of bonds but does not baseline erode partner preference

We measured partner-directed affiliation via a partner preference test (PPT) across time in animals that remained paired and before and after short- or long-term partner separation (*Figure 1A*). The PPT measures the amount of time spent huddling either with the partner or a novel vole tethered at opposite ends of an arena (*Figure 1A*, right side). To ensure that the experimental males could not hear, smell, or see their separated partners during separation, all partners were moved to a separate vivarium room. All sample sizes and comprehensive statistical results, including effect size estimates, are reported in *Supplementary file 1*.

We initially hypothesized that opposite-sex pairs would have a partner preference at baseline and after short-term separation, when the pair bond is still intact, but not after long-term separation (*Harbert et al., 2020*; *Sun et al., 2014*). Similarly, we anticipated that same-sex sibling pairs would have a partner preference at baseline due to familiarity but would exhibit a faster and more robust loss of partner preference when separated (*Lee et al., 2019*).

In contrast to our initial hypothesis, we found that opposite-sex and same-sex paired males demonstrated a partner preference even after extended separation (*Figure 1B*; *Lee et al., 2019*; *Goodwin et al., 2019*). In pairs that remained with their partners throughout the 6-week timeline, both opposite-sex and same-sex pairs form a selective preference for their partner after 2 weeks of pairing (*Figure 1B*, one sample t-test relative to 50%: opposite-sex p=0.00564, same–sex p=0.00922). The baseline partner preference for both opposite- and same-sex pairs is retained—and even moderately strengthened—after 4 additional weeks of cohabitation (one sample t-test relative to 50% for 48 hr PPT: opposite-sex p=0.0107, same-sex p=0.01478; for 4 weeks PPT: opposite-sex p=5.19 × 10$^{-7}$, same-sex p=6.49 × 10$^{-7}$, two-way RM-ANOVA in *Supplementary file 1*). In separated animals, both opposite-sex and same-sex pairs form a selective preference for their partner after 2 weeks of cohabitation (one sample t-test relative to 50% short-term separated animals: opposite-sex p=3.30 × 10$^{-12}$, same-sex p=7.72 × 10$^{-5}$; long-term separated animals: opposite-sex p=1.50 × 10$^{-3}$, same-sex p=3.04 × 10$^{-5}$) and retain their partner preference after short-term and long-term separation (one sample t-test relative to 50%: short-term separated animals: opposite-sex p=4.10 × 10$^{-7}$, same-sex p=3.66 × 10$^{-8}$; long-term separated animals: opposite-sex p=0.001846, same-sex p=3.14 × 10$^{-8}$). Additionally, there was no significant difference in partner preference score due to either separation duration or partner type (*Figure 1B*; two-way RM-ANOVA with repeated measures for timepoint: short-term separation: Main effect of partner (opposite- vs same-sex): $F_{(1,28)}$ = 0.38, p=0.54; Main effect of time: $F_{(1,28)}$ = 0.22, p=0.64; Partner X time: $F_{(1,28)}$=1.87, p=0.18; long-term separation: Main effect of partner (opposite- vs same-sex): $F_{(1,26)}$ = 1.70, p=0.20; Main effect of time: $F_{(1,26)}$ = 0.16, p=0.70; Partner X time: $F_{(1,26)}$=0.057, p=0.81). Together, these results indicate that prairie voles are capable of forming an affiliative preference for an opposite- or same-sex partner and that preference remains intact despite prolonged separation.

Next, we examined how partner- and novel-directed affiliative behavior, respectively, changed over time. For males that remain paired there were no significant differences in novel huddle over time for either pairing type. Of note, partner huddle increases with extended co-housing in both pairing

types indicting a strengthening of partner-directed affiliation (2-way RM-ANOVA: main effect of time-point (4 wks:base): $F_{(2, 42)}$=4.92, p=0.012, $\eta$=0.170) (*Figure 1C*). This increase was not observed in opposite-sex or same-sex paired males that had been separated from their partner (*Figure 1D and E*; two-way RM-ANOVA in *Supplementary file 1*). This suggests that long-term separation effectively impedes the maturation and strengthening of the pair bond without necessarily dissolving the baseline preference. We also quantified how long it took to begin huddling in the partner preference test but failed to observe any consistent differences for partner or novel huddle latencies over time regardless of experimental condition or pairing type (*Figure 1—figure supplement 1A–F*, log-rank test in *Supplementary file 1*).

Additionally, we quantified how much each animal's partner preference and partner huddle changed between timepoints (endpoint minus baseline; *Figure 1—figure supplement 2A, B*, stats in *Supplementary file 1*). These change scores revealed that partner preference and partner huddle were relatively stable between the baseline and short-term timepoint regardless pairing status or pairing type. However, at the long-term timepoint, we observed another indicator that separation impedes ongoing pair bond strengthening: opposite-sex males that remain paired exhibited increased partner huddle duration at 4 weeks relative to baseline, which was not observed in opposite-sex separated males (*Figure 1—figure supplement 2B*). In contrast, in same sex paired males, ongoing pairing was not required for increased partner huddle at the long-term timepoint (*Figure 1—figure supplement 2B*).

Finally, we assessed behavioral consistency across PPTs by determining the correlation of partner preference test metrics (huddle times, partner preference score, and chamber times) between baseline to short-term and baseline to long-term timepoints in all cohorts (*Figure 1—figure supplement 3*). Pairing duration, separation, and partner type did not yield reliable differences in correlated patterns of behavior. Together, these results indicate that prairie voles form an affiliative preference for an opposite- or same-sex partner that remains intact despite prolonged separation. However, separation impedes the normal, strengthening of partner-directed affiliation that occurs over time.

## The pair bond transcriptional signature is stable in intact bonds

Changes in behavioral states, such as a shift in dominance or reproductive status, are supported by stable changes in transcription, although this has not been examined in the context of pair bonds (*Cardoso et al., 2015*; *Zayed and Robinson, 2012*; *Tripp et al., 2018*). While prior work has demonstrated that mating and cohabitation in prairie voles result in transcriptional changes within the NAc, the consistency of these changes as long as the bond remains intact has yet to be assessed (*Duclot et al., 2020*; *Tripp et al., 2021*; *Resendez et al., 2016*; *Resendez et al., 2012*; *Aragona et al., 2006*). Thus, we compared NAc transcription in opposite- versus same-sex-paired voles following either 2 or 6 weeks of pairing/cohabitation (*Figure 2A*). By comparing opposite-sex to same-sex-paired animals, we identified transcripts specific to pair bonds compared with those associated with affiliative behavior more generally. For consistency, we limited transcriptional assessment to voles that had a baseline partner preference >50% (*Figure 2B*; *Supplementary file 1*; OS n=15; SS n=11. Excluded for PPT <50% OS n=3; SS n=8).

We used DESeq2 to identify transcripts up- or down-regulated in opposite- relative to same-sex paired males after short-term and long-term pairing (*Figure 3—figure supplement 2A, B*; *Love et al., 2014*). When ordering transcripts based on their log$_2$FoldChange at the short-term timepoint, the global transcriptional differences for opposite- versus same-sex pairs were strikingly similar following either 2 or 6 weeks of pairing, suggesting stable pair bond associated transcription across these time-points (*Figure 2D*; Spearman's Rho = 0.38, p=2.2 × 10$^{-16}$). We further determined that the observed correlation across timepoints is stronger than what would be expected by chance. We shuffled each animal's cohort identity at the long-term timepoint and calculated Rho values of the log$_2$FoldChange of differential gene expression between the observed short-term pair bond and the shuffled long-term pair bond over 1000 iterations (*Figure 2—figure supplement 1A–D*). The cohort identity is a combination of three variables: pairing type (opposite-sex or same-sex), separation condition (remain paired or separated), and timepoint (short- or long-term). For example, long-term separated opposite-sex animals is one cohort identity of 8 total. This approach effectively randomized timepoint, partner type, and pairing status without disrupting underlying structure that exists within our transcriptional dataset due to correlations in expression across genes. Our observed Rho value was greater than

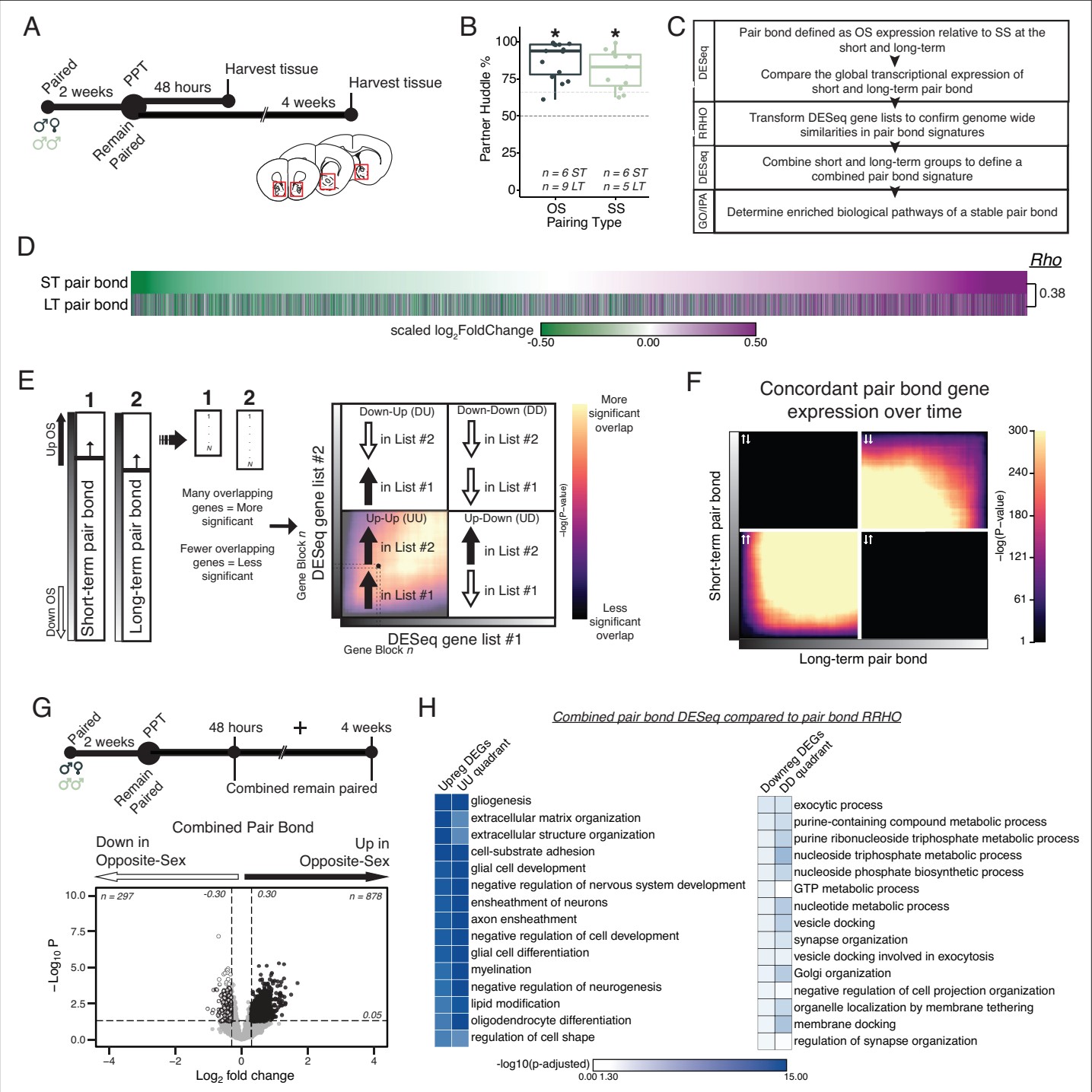

**Figure 2.** Pair bonding leads to persistent and consistent changes in NAc transcription. (**A**) Opposite- and same-sex pairs were paired for 2 weeks prior to a baseline partner preference test. Pairs then remain paired for either 48 hours (short-term; ~2 weeks total pairing) or 4 weeks (long-term; ~6 weeks total pairing) prior to collecting fresh nucleus accumbens tissue (dissection sites in red boxes) for RNA sequencing. (**B**) Baseline partner preference scores of males included in RNA sequencing for the opposite- and same-sex groups (one-tailed t-test relative to 50%: opposite-sex $T_{14}$=11.76, p=1.21 X $10^{-8}$; same-sex $T_{10}$=7.78, p=1.50 X $10^{-5}$). Black dotted line indicates a 50% partner preference score and the grey dotted line indicates 66%. There were no differences in partner preference score between opposite- and same-sex paired animals used for RNAseq (two-tailed t-test: $T_{21.016}$ = 1.374, p=0.184). (**C**) Transcriptional analysis workflow. (**D**) Gene list from both timepoints ordered from the smallest to largest log$_2$FoldChange after short-term pairing with color indicating up- or down-regulation in opposite- vs same-sex pairs. Expression patterns are strongly correlated across timepoints (Rho = 0.38, p=2.2 X $10^{-16}$). (**E**) Schematic of RRHO analysis. The heatmap is arranged into quadrants of genes upregulated in both lists (up-up: quadrant UU),

*Figure 2 continued on next page*

*Figure 2 continued*

downregulated in both lists (down-down: quadrant DD), or genes that have opposite regulation (up in list 1-down in list 2: quadrant UD; down in list 1-up in list 2: quadrant DU). Genes that are found in both lists at a similar ranked position result in higher p-values and are represented by a yellow color. (**F**) RRHO comparing short-term and long-term pair bonding (*from 2D*) indicates a stable pair bond gene signature over time as evidenced by concordant up- or downregulated genes at the two timepoints. (**G**) The short-term and long-term time points were pooled for opposite- and same-sex pairs to define the combined pair bond gene signature. (**H**) We compared GO analysis *Mus musculus* ontology terms between the combined pair bond DEGs and the RRHO quadrants (*from 2 F*) with strong correspondence between the two analyses.

The online version of this article includes the following figure supplement(s) for figure 2:

**Figure supplement 1.** Cohort shuffled controls for remain paired cohorts.

**Figure supplement 2.** Ingenuity Pathway Analysis (IPA) corresponding to GO term analyses.

100% of iterations, indicating that the similarity in gene expression across short and long-term pairing timepoints is greater than would be expected by chance (*Figure 2—figure supplement 1D*). Next, we asked whether our observed association across timepoints was driven by moderately but consistently expressed genes. Eliminating transcripts whose expression difference fell within the middle two quartiles in the short-term group resulted in a greater Rho value ($R=0.44$, $p<2.2 \times 10^{-16}$), indicating that the correlation across short- and long-term timepoints is more strongly driven by transcripts with larger expression differences between opposite- and same-sex pairs. To further interrogate the biological function of altered transcripts, we set nominal thresholds of $log_2FoldChange >0.30$ or$<-0.30$, which are sufficient to identify replicable differences detectable via alternate methods (*Walker et al., 2018*), along with a p-value threshold of $p<0.05$ to identify up- and down-regulated gene lists (DEGs). This nominal p-value threshold was used with the aim of identifying gene families regulating molecular pathways responsive to bonding instead of specific gene candidates. Prior studies in animal brain tissue using this p-value have yielded biologically meaningful insights (*Walker et al., 2018*; *Mukamel, 2022*; *Kronman et al., 2021*; *Walker et al., 2022*; *Labonté et al., 2017*; *Krishnan et al., 2007*; *Peña et al., 2017*; *Seney et al., 2018*; DEGs based on these thresholds are listed in *Supplementary file 2*). There was significant overlap in the upregulated (104 shared, Fisher's Exact Test: $\chi2=30.24$, $p=5.36 \times 10^{-31}$) and downregulated (46 shared, Fisher's Exact Test: $\chi2=13.32$, $p=7.05 \times 10^{-14}$) transcripts across timepoints (*Figure 3—figure supplement 2E*; *Wang et al., 2015*).

To further examine similarity in transcriptional patterns between short-term and long-term pair bonds we employed Rank-Rank Hypergeometric Overlap (RRHO; *Figure 2E*; *Plaisier et al., 2010*; *Cahill et al., 2018*). This threshold free approach allows us to determine if similar global transcription patterns are observed between two comparisons. The resulting RRHO heatmap is arranged into quadrants based on the direction of gene expression and each point represents the significance derived from the number of overlapping genes via the hypergeometric distribution (*Figure 2E*). We determined how similar global transcription is between short-term and long-term pair bonds by comparing differential gene expression in the opposite- vs same-sex 2-week paired to opposite-sex vs same-sex 6-week paired groups. We observed extensive concordance of transcriptional patterns between 2- and 6-week paired voles, further supporting that the transcriptional profile of pair bonds is stable (*Figure 2F*). Confirming that our observed RRHO signal was unlikely to be attributable to chance, we randomly shuffled cohort identity (*Figure 3—figure supplement 4E*) and, separately, shuffled the gene ranking of the combined pair bond genes (*Figure 3—figure supplement 4G*), both of which ablated the RRHO signal. Since the pair bond transcriptional profiles are concordant at both time points, the 2-week and 6-week animals from each pairing type were pooled to create a single, well-powered opposite-sex- vs same-sex-paired comparison that defines the combined pair bond transcriptional signature (*Figure 2G*).

To determine which biological pathways might underlie pair bonding we employed Gene Ontology (GO) and Ingenuity Pathway Analysis (IPA) (*Yu et al., 2012*). Gene lists for GO terms or IPA were generated either from DEG lists or from genes in concordant (UU or DD) RRHO quadrants. GO terms that passed a false discovery rate of $p<0.05$ were retained while IPA terms that had activation scores of at least –2 or 2 and whose enrichment for genes predicted to be regulated was significant (p-value $<0.05$) were kept (*Supplementary file 3*). Combined pair bond GO terms associated with the upregulated DEGs and the quadrant UU genes strongly implicate changes in glial cells and extracellular matrix organization (*Figure 2H*), which is mirrored by activation of glioblastoma signaling via IPA (*Figure 2—figure supplement 2A*). IPA additionally identified upregulation of Synaptogenesis, CREB

signaling, and Endocannabinoid Neuronal Synapse pathways. All these pathways are consistent with pair bonding as a form of complex learning that is mediated by neuromodulatory signaling (*Walum and Young, 2018*; *Loth and Donaldson, 2021*; *Modi and Young, 2011*; *Johnson and Young, 2017*). Predicted upstream regulators of these pathways broadly support a role for learning and neuromodulation, including Creb1, Estrogen receptor alpha (Esr1), and various growth-factor and developmental genes (*Figure 2—figure supplement 2A*). Combined pair bond GO terms associated with the downregulated DEGs and the quadrant DD genes are implicated in nucleoside synthesis and synaptic plasticity (*Figure 2H*). IPA indicates a suppression of corticotropin releasing hormone signaling, potentially reflecting the strong social buffering of stressors that occurs in pair bonded voles specifically (*Bosch et al., 2016*; *Smith et al., 2013*; *Bosch et al., 2009*; *Smith and Wang, 2014*; *Burkett et al., 2016*; *Figure 2—figure supplement 2A*). Finally, the strong correspondence in the significance of GO terms between the combined pair bond DEGs and the RRHO quadrants further validates that pooling the time points to represent a single pair bond transcriptional signature retains biologically relevant information (*Figure 2H*).

## The pair bond transcriptional signature erodes following prolonged partner separation

We next tested the hypothesis that the pair bond transcriptional signature erodes following long-term partner separation. We paired opposite-sex and same-sex pairs for 2 weeks prior to a baseline partner preference test and then immediately separated males into new cages where they were singly housed for either 48 hours (short-term) or 4 weeks (long-term) prior to harvesting NAc tissue for RNAseq (*Figure 3A*). As before, we only included animals with a baseline partner preference >50% (*Figure 3B*; OS n=12; SS n=11. Excluded OS n=4; SS n=5). We determined the effect of separation on the pair bond gene signature by first defining differential gene expression of opposite- vs same-sex animals after short- and long-term separation and then used RRHO to examine genome-wide expression changes relative to the combined pair bond signature (*Figure 3C*).

First, we performed differential expression analysis of opposite- vs same-sex-separated males at each separation time point (*Figure 3—figure supplement 2C and D*). There were significantly more shared DEGs between the combined pair bond group and short-term separation group than would be expected by chance, but this was not observed when comparing the shared genes between the combined pair bond and long-term separation group (*Figure 3—figure supplement 2F* Fisher's exact test: pair bond:short-term sep $\chi2=31.86$, $p=6.78 \times 10^{-19}$; pair bond:long-term sep $\chi2=17.96$, $p=0.53$). The difference in transcription between short- and long-term separation is particularly evident when comparing the global transcriptomes through the lens of pair bond transcription. Specifically, when genes are ordered based on their degree of down- or up-regulation in the pair bond group we found that gene expression was largely indistinguishable in short-term separated animals (Rho = 0.32, $p<2.2 \times 10^{-16}$) and distinct from long-term separated animals (Rho = 0.048, $p=1.6 \times 10^{-7}$), indicating that the pair bond signature is largely eroded after prolonged separation (*Figure 3D*). We further assessed whether these correlations differed from chance via the previously employed permutation analysis. We found that the pair bond:short-term Rho value indicated a stronger association than would be expected by chance (p=0.0362) while the pair bond:long term separation association value was highly likely to occur by chance (p=0.41). Together, these results suggest that extended separation erodes nucleus accumbens pair bond transcription, a potentially crucial molecular process needed to adapt to partner loss.

To more comprehensively assess genome-wide changes to the pair bond signature following extended partner separation we again employed RRHO (*Figure 3E*). We compared the ranked gene list of opposite- vs same-sex for the (1) combined pair bond gene signature (*Figure 3D*, top) versus either (2) short-term partner separation (*Figure 3D*, middle) or (3) long-term partner separation (*Figure 3D*, bottom). We found that the pair bond signature remains largely concordant after short-term separation as indicated by significant signal among upregulated genes (quadrant UU), and to a lesser extent, downregulated genes (quadrant DD; *Figure 3E*). However, following long-term separation, the pair bond gene signature is largely undetectable as indicated by a lack of significant signal throughout the RRHO plot. Plotting the long-term RRHO with an unadjusted p-value scale revealed few overlapping genes in the UU and DD quadrants (scale 1–16 *Figure 3—figure supplement 4C* rather than 1–300 as in *Figure 3D*). The low level of overlap indicates that there is a residual, albeit

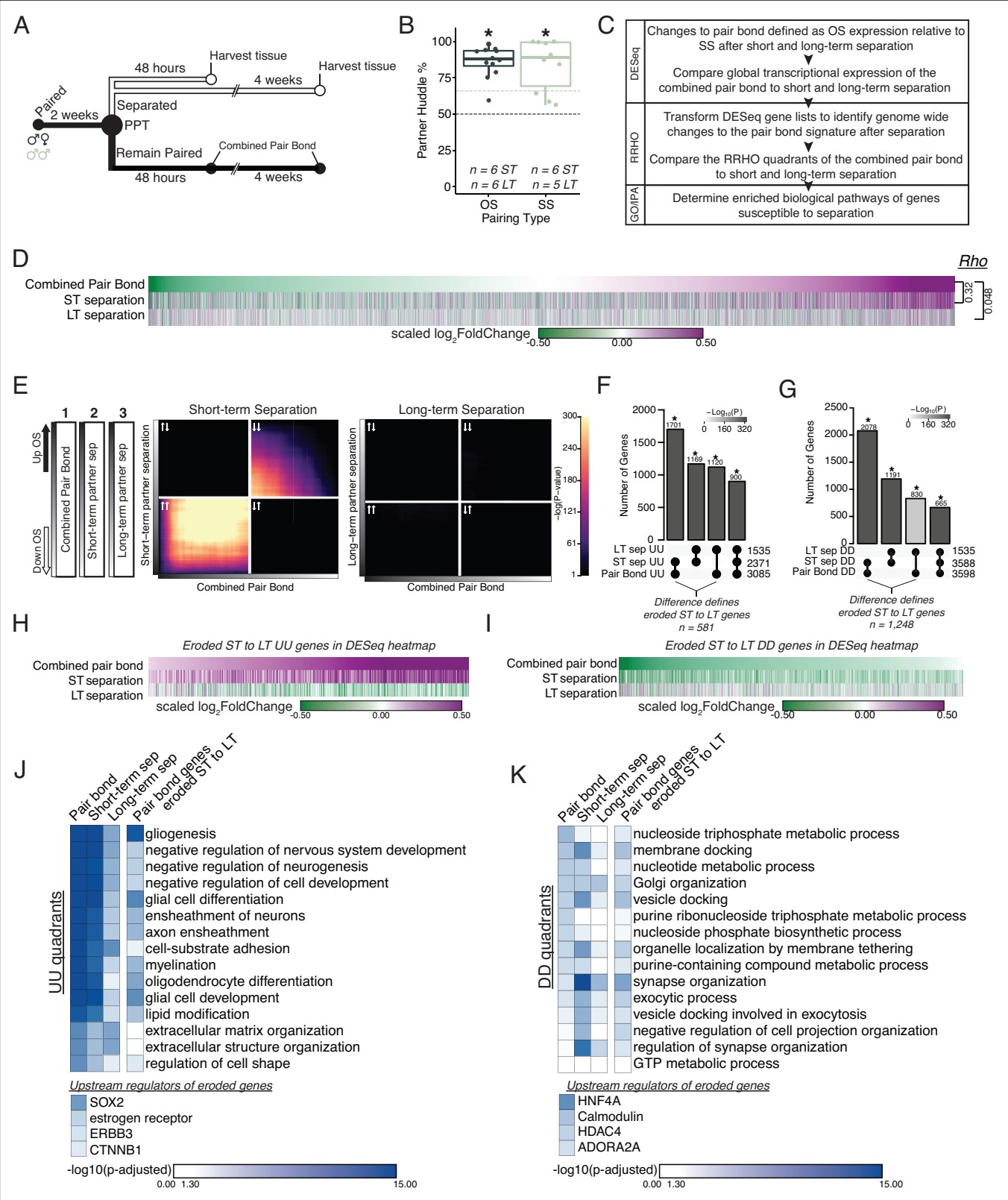

**Figure 3.** Prolonged separation erodes pair bond transcriptional signatures. (**A**) Opposite-sex and same-sex pairs were paired for 2 weeks prior to a baseline partner preference test. Pairs were then separated for either 48 hours (short-term) or 4 weeks (long-term) prior to collecting fresh nucleus accumbens tissue for RNAseq. Animals in the Remain Paired group are analyzed in *Figure 2* to define the Combined Pair Bond transcriptional signature. (**B**) Baseline partner preference scores of males included in RNA sequencing for the opposite- and same-sex separation groups (one-tailed t-test relative

*Figure 3 continued on next page*

*Figure 3 continued*

to 50%: opposite-sex $T_{11}$=11.56, p=1.71 X $10^{-7}$; same-sex $T_9$=6.07, p=1.87 X $10^{-4}$). Black dotted line indicates a 50% partner preference score and the grey dotted line indicates 66%. There were no differences in partner preference score between opposite-sex and same-sex separated animals used for RNAseq (two-tailed t-test: $T_{14.33}$ = 0.33, p=0.75). (**C**) Transcriptional analysis workflow similar to *Figure 2*. (**D**) Heatmap of the scaled log$_2$FoldChange for every gene where short- and long-term separation is compared to the combined pair bond gene signature. (**E**) RRHO using the combined pair bond, short-term separated, and long-term separated ranked transcript lists show that short-term separated animals retain a gene expression pattern concordant with pair bond transcription while there is a dramatic erosion of pair bond gene expression following long-term separation. (**F, G**) Upset plot showing overlap of genes in the UU or DD quadrants of *2F* and *3E* RRHO was determined using a Fisher's Exact test. The difference between the intersections of the pair-bond:short-term sep. and pair-bond:long-term sep. was used to define the eroded gene lists. (**H, I**) The RRHO gene lists of eroded pair bond genes were used to filter the DESeq heatmap in *2D*. Eroded genes from the UU quadrants show upregulation during pair bonding and short-term separation but downregulation after long-term separation (**H**) while the eroded genes from the DD quadrants show the opposite pattern (**I**). (**J, K**) GO and IPA analysis of RRHO quadrant gene lists and pair bond eroded gene lists. Scale represents the –log$_{10}$(p-adjusted) where any non-significant terms are white. (**J**) Glial associated GO terms of the UU quadrants are less significantly represented after separation and are highly significantly represented among the eroded genes. Significant upstream regulators of the eroded genes include Sox2, estrogen receptor, Erbb3, and Ctnnb1. (**K**) Vesicle docking and synapse organization associated GO terms of the DD quadrants are less significantly represented after separation and are more significantly represented among the pair bond eroded genes. Significant upstream regulators of the eroded genes include Hnf4a, Calmodulin, Hdac4, and Adora2a.

The online version of this article includes the following figure supplement(s) for figure 3:

**Figure supplement 1.** Cohort shuffled controls for the combined pair bond and separation time points.

**Figure supplement 2.** Volcano plots of differential expression with Upset plots for overlap of differential expressed genes.

**Figure supplement 3.** Randomly selected subsampled data shows the same differential expression and RRHO patterns as the full dataset.

**Figure supplement 4.** Control analyses for RRHO.

**Figure supplement 5.** RRHO of short-term vs long-term partner separation reveals a transcriptional signature of partner loss.

greatly reduced, pair bond signature that is still detectable after long-term separation (***Figure 3— figure supplement 4C***). Additionally, we used the same control analyses as previously employed (***Figure 3—figure supplement 4G–I***).

We next examined which genes are consistently present in the UU or DD quadrants between pair bonding and each separation timepoint. Specifically, we compared the UU and DD quadrants (inclusive of all transcripts in each quadrant) of the short-term (2 week) vs long-term (6 week) pair bond from ***Figure 2F***, the combined pair bond vs short-term separation in the left side of ***Figure 3E***, and the combined pair bond vs long-term separation on the right side of ***Figure 3E***. The degree of overlap in the UU quadrants was greatest between the combined pair bond vs short-term separation and least between the combined pair bond vs long-term separation, a result consistent with the most extensive erosion of pair bond transcription occurring after long-term separation (***Figure 3F***: all stats in ***Supplementary file 1***, genes in ***Supplementary file 2***). The same trend was observed for transcripts in the DD quadrants (***Figure 3G***, all stats in ***Supplementary file 1***, genes in ***Supplementary file 2***). We next identified genes whose differential expression in pair bonded voles eroded as a function of separation time. We reasoned that such genes are involved in biological pathways that likely regulate adapting to partner loss and prime an animal to form a new pair bond. The genes that were no longer present in the pair bond:long-term separation intersection that were present in the pair bond:short-term separation intersection were deemed eroded genes. These eroded transcripts include 581 UU quadrant transcripts and 1,248 DD quadrant transcripts (***Figure 3F and G***). We next filtered the genome-wide expression data of ***Figure 3F*** by the eroded gene lists (***Figure 3H,I***). The eroded UU genes are strongly upregulated in a stable pair bond and show dramatic downregulation following long-term separation (***Figure 3H***). Thus, the eroded UU genes and their associated biological processes may represent key aspects of adapting to partner loss. There was also a corresponding reduction in DD gene overlap between a stable pair bond and following separation, further supporting that bond-related changes in gene expression erode after long-term separation regardless the initial direction of expression during bonding (***Figure 3I***).

We next determined how separation-induced transcriptional changes affected pair bond associated biological processes. We used the pair bond associated GO terms from ***Figure 2H*** as a reference point to determine if these terms were still significantly associated with the underlying transcriptional profile following partner separation (***Figure 3J and K***). We predicated two outcomes. First, separation would reduce the significance of the pair bond associated GO terms and second, pair bond genes

whose expression eroded would be significantly represented by the original pair bond associated terms. The transcripts in the short-term separation UU quadrant were still significantly associated, though to a lesser degree, with the pair bond associated GO terms. After long-term separation, the underlying transcriptional signature is much less significantly associated with the pair bond GO terms. Further, the eroded transcripts are highly associated with the pair bond associated GO terms, including glial-associated terms. A potential role for glia was also implicated in the IPA identification of pathways for glioblastoma multiforme signaling and glioma signaling (*Figure 2—figure supplement 2C*). Upstream regulators inducing the erosion of these glial-associated genes include Sox2, estrogen receptor, Erbb3, and Ctnnb1 (*Figure 3J*, *Figure 2—figure supplement 2C*). We observe a similar trend in the DD quadrants where the pair bond associated GO terms become less representative of the underlying transcriptional signature following long-term separation but are associated with the eroded genes, although to a lesser degree (*Figure 3K*). IPA pathways activated in a stable pair bond (endocannabinoid signaling, synaptogenesis, and dopamine signaling) are significantly represented among the eroded DD genes (*Figure 2—figure supplement 2C*). Upstream regulators for the eroded DD transcripts include Hnf4a, Calmodulin, Hdac4, and Adora2a (*Figure 3K*, *Figure 2—figure supplement 2C*).

Finally, to gain more nuanced insight into the temporal trajectory of erosion, we also used RRHO to compare short-term to long-term partner separation (*Figure 3—figure supplement 5*). In contrast to the pattern of consistently upregulated genes between the pair bond and short-term separation, we observed high concordance of downregulated genes between separation timepoints (*Figure 3—figure supplement 5A*). We then identified transcripts in the UU and DD quadrants of the separation RRHO that exhibited the largest inversion of expression between pair bond and short-term separation (details in Materials and Methods). We reasoned that such transcripts represented potential gene sets that are rapidly eroded following partner separation and which remain eroded after long-term separation, representing a potential first wave transcriptional response to partner loss. We further asked which biological processes are represented by these consistently inverted genes via GO analysis (*Figure 3—figure supplement 5B, C*). We found that UU pair bond genes that are downregulated following pair bond separation are associated with dopaminergic processes and DD pair bond genes that are upregulated are associated with neuronal rearrangement processes. Thus, this RRHO analysis enriches our interpretation of transcriptional erosion of pair bond signatures following partner separation by revealing temporally complex shifts in gene expression resulting from partner loss.

## Neuronal-specific interrogation reveals gene clusters associated with pair bond disruption and loss adaptation

Our tissue-level analysis of gene expression strongly implicated a role for glia in bond formation and loss, and we reasoned that relevant neuronal transcriptional changes may have been masked by these prominent glial transcriptional signals. Thus, we specifically examined separation-induced neuronal transcriptional changes by pioneering translating ribosome affinity purification in voles (vTRAP) (*Heiman et al., 2014*; *Heiman et al., 2008*). The vTRAP construct, DIO-eGFP-pvRPL10a, is a double-floxed inverse orientation eGFP-tagged ribosomal subunit that uses the prairie vole RPL10a gene sequence (*Figure 4A*; *Figure 4—figure supplement 1A*). The vTRAP construct is paired with Cre-recombinase (Cre) for inducible, cell-type specific expression. AAV-mediated delivery of hSyn-Cre, which drives neuron-specific Cre-recombinase expression, and hSyn-DIO-vTRAP vectors results in neuronal expression of GFP-tagged ribosomes. Subsequent immunoprecipitation of the tagged ribosomes provides a means to isolate neuron-specific, actively-translating mRNAs (*Heiman et al., 2014*; *Heiman et al., 2008*). vTRAP is Cre-dependent as confirmed by bilateral NAc injections of +/-Cre (*Figure 4—figure supplement 1A*). We performed vTRAP immunoprecipitation of 3 cohorts: (1) opposite-sex long-term separated, (2) opposite-sex long-term remain paired, and (3) same-sex long-term separated.

We defined a set of neuronally enriched genes by using the intersection of threshold and threshold-free based analyses. We first identified neuronally-enriched transcripts by comparing the transcriptional profiles of the input fraction (equivalent to bulk RNA-seq) and pulldown fraction of the animals from the TRAP groups specified above. We predicted that the pulldown fraction would be enriched for neuronal markers and depleted in glial-associated transcripts. Differential expression analysis using the previously defined thresholds revealed that most transcripts are depleted in the pulldown

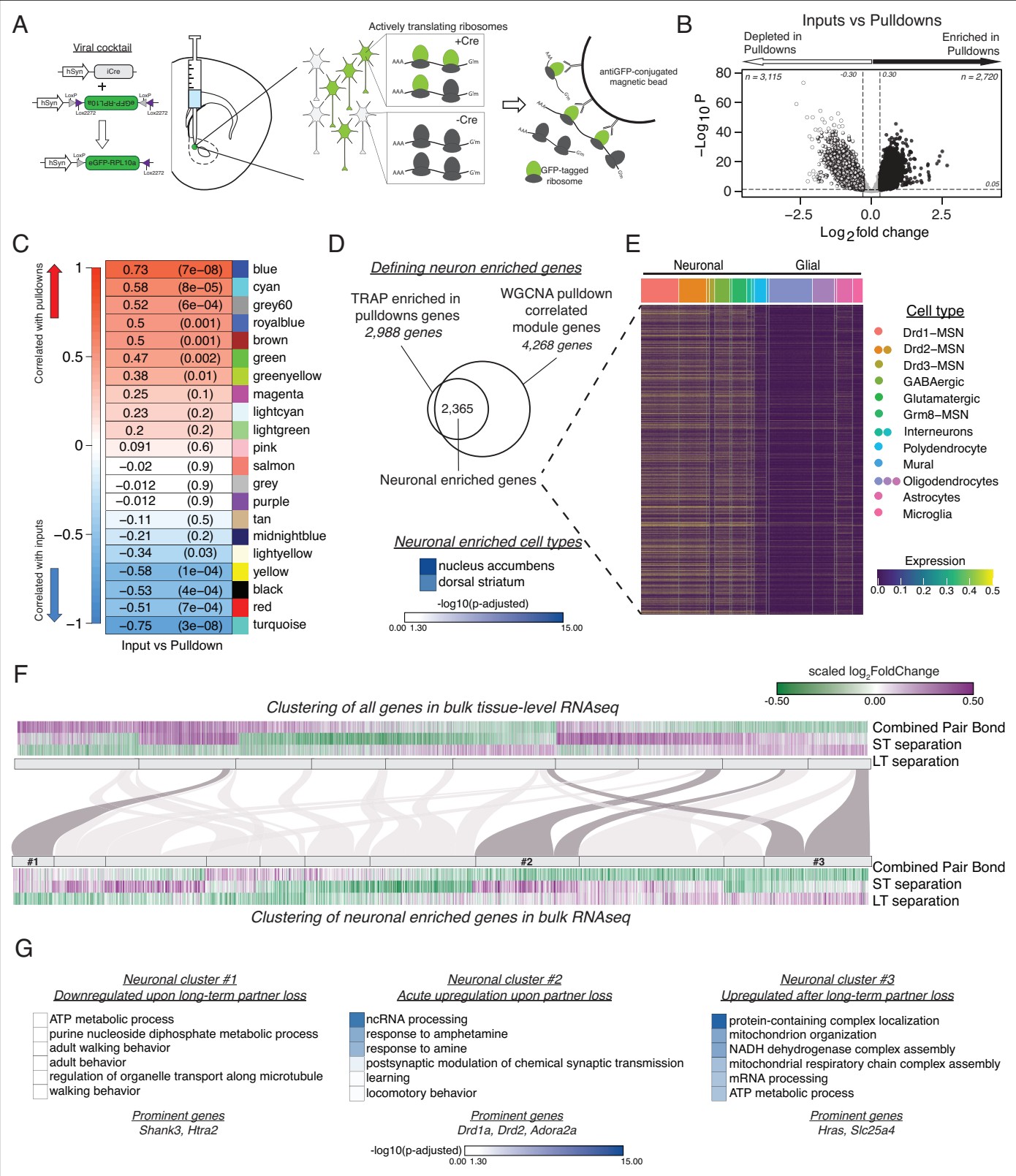

**Figure 4.** vTRAP elucidates separation-induced transcriptional changes specifically in neurons. (**A**) Schematic of the vTRAP system. AAV-mediated delivery of hSyn-Cre, which drives neuron-specific Cre-recombinase expression, and DIO-vTRAP vectors results in neuronal expression of GFP-tagged ribosomes. Subsequent immunoprecipitation of the tagged ribosomes isolates neuron-specific, actively-translating mRNAs. (**B**) Transcripts enriched in the immunoprecipitation pulldowns compared to the input fractions. Differential expression skews towards depletion. (**C**) WGCNA to identify gene

*Figure 4 continued on next page*

*Figure 4 continued*

modules correlated with the input (negative correlation) or pulldown (positive correlation) fractions. The grey60 module contains neuronal transcripts of interest such as Drd1a, Fosb, Oprk1, and Pdyn while the blue module contains Drd2. (**D**) Neuronal enriched genes—2,365 genes—were defined as the intersection of the Input v Pulldown enriched DEGs and the WGCNA pulldown-correlated modules (modules grey60, blue, greenyellow, brown, green, royalblue, and cyan). When using Enrichr to search for cell types associated with the neuronal enriched gene list, nucleus accumbens is the most prominent and is highly significant followed by dorsal striatum. Scale represents –$\log_{10}$(p-adjusted). (**E**) Heatmap showing expression of identified neuronally enriched genes in a single-nuclei RNAseq dataset from the nucleus accumbens of adult male rats. The scale indicates expression of each gene in the snRNAseq dataset. (**F**) Hierarchical clustering using one minus Pearson correlation with complete linkage of bulk tissue-level sequencing for all genes (top heatmap) and neuronal enriched genes (bottom heatmap) for the combined pair bond, short-term separation, and long-term separation. The scale indicates scaled $\log_2$FoldChange. The mapping of neuronal clusters to the all genes clusters is visualized via a Sankey plot. Neuronal clusters of interest are highlighted by dark grey links. (**G**) Gene Ontology analysis using mouse terms for each neuronal cluster of interest. Genes of interest that occur in multiple terms are referred to as prominent genes. Scale represents –$\log_{10}$(p-adjusted).

The online version of this article includes the following figure supplement(s) for figure 4:

**Figure supplement 1.** Alignment of mouse (*Mus musculus*) and prairie vole (*Microtus ochrogaster*) ribosomal protein L10a (RPL10a) cDNA sequences and amino acid sequences.

**Figure supplement 2.** Validation of expression and analyses for vole optimized Translating Ribosome Affinity Purification.

fraction as expected (*Figure 4B*). GO terms support that hSyn-vTRAP successfully isolates neuronally-enriched transcripts and filters out genes associated with gliogenesis (*Figure 4—figure supplement 2C*). Second, we used weighted correlation network analysis (WGCNA) to identify gene modules significantly correlated with the input (negative correlation) or pulldown (positive correlation) fractions (*Figure 4C*, *Figure 4—figure supplement 2B*; *Langfelder and Horvath, 2008*). Of note, two of the most significantly positively correlated modules, blue and grey60, contain multiple neuronal genes of interest such as Drd1a, Drd2, Oprk1, Fosb, and Pdyn, which have previously been implicated in pair bonding and reward learning (*Resendez et al., 2016*; *Resendez et al., 2012*; *Pitchers et al., 2010*). Enrichr analysis for cell type also indicated that our identified neuronal enriched genes are predominantly associated with the nucleus accumbens and dorsal striatum (*Figure 4D*; *Chen et al., 2013*; *Kuleshov et al., 2016*; *Xie et al., 2021*). Finally, we validated that the gene list at the intersection of the DEGs and the WGCNA modules was predominately expressed in neurons and not glia using a publically available single nuclei NAc gene expression dataset generated from adult male rats (*Savell et al., 2020*). We filtered the dataset by our neuronal enriched gene list and plotted those gene's expression in the rat nucleus accumbens, which confirmed that our genes are predominately expressed in neuronal cell types (*Figure 4E*).

We next queried potential neuronal transcriptional changes in our tissue-level RNAseq data. To ensure that the tissue-level expression is representative of neuron-specific expression we compared the expression patterns of the neuronal enriched genes in the input and pulldown fractions of opposite- vs same-sex long-term separated males (*Figure 4—figure supplement 2D*). The strong similarity in expression validates that bulk sequencing expression can be used to infer the neuronal expression of those same genes (*Figure 4—figure supplement 2D*). As such, we filtered the tissue-level RNAseq dataset from *Figure 3D* by the neuronal enriched gene list of *Figure 4D* to track neuronal gene expression throughout partner loss. We then performed unsupervised clustering of all genes (*Figure 4F*, top) and neuronal enriched genes (*Figure 4F*, bottom). We identified three neuronal-enriched gene clusters that had particularly interesting expression patterns. Cluster 1 consists of genes that were upregulated in opposite-sex paired males during pair bonding and short-term separation and downregulated following long-term separation. GO terms associated with these genes include adult walking behavior, a term that contains the autism-associated gene Shank3 (*Figure 4G*; *Gong et al., 2015*; *Durand et al., 2007*). Cluster 2 transcripts are downregulated in pair bonded males and then robustly upregulated after short-term partner loss and, to a lesser extent, at the long-term timepoint, indicating an acute transcriptional response to pair bond disruption. The GO terms associated with these genes suggest an involvement of dopaminergic systems (Drd1a and Drd2) and learning processes potentially engaged to adapt to loss (*Figure 4G*). Finally, Cluster 3 contains genes that were downregulated in pair bonded males and became upregulated only after long-term opposite-sex partner separation, representing transcripts that may help prime the vole to form a new bond. The GO terms associated with this cluster are primarily implicated in cellular metabolism (*Figure 4G*). IPA revealed that neuronal clusters 2 and 3 were also enriched for estrogen receptor

signaling and oxidative phosphorylation pathways with cluster 2 also implicating opioid signaling pathways (*Figure 2—figure supplement 2D*). Steroid and opioid signaling have been extensively implicated in pair bonding (*Resendez et al., 2016*; *Resendez et al., 2012*; *Loth and Donaldson, 2021*; *Inoue et al., 2013*; *Stetzik et al., 2018*; *Carter and Perkeybile, 2018*; *Lei et al., 2010*; *Cushing et al., 2008*).

## Discussion

Our data provide the first comprehensive assessment of social behavior and accumbal transcription in pair bonded male prairie voles before and after partner loss, providing novel insight into the biological processes that may enable loss adaptation. Prairie voles, unlike laboratory mice and rats, form exclusive lifelong pair bonds that are distinct from affiliative same-sex relationships, making them ideal for studying the unique neurobiology of bonding and loss (*Getz et al., 1981*; *Carter et al., 1995*; *Pohl et al., 2019*; *Osako et al., 2018*; *Lee et al., 2019*; *Goodwin et al., 2019*; *Lee and Beery, 2021*). Additionally, the direct comparison of opposite- versus same-sex paired animals provides an important control for the effects of social context – whether living with another animal or alone (*Arzate-Mejía et al., 2020*; *Wallace et al., 2009*). By leveraging this comparison across our experiments, we demonstrated that pair bonding leads to consistent and persistent shifts in NAc transcription, which erode as a function of long-term partner separation. This suggests that substantial changes in accumbal gene expression may contribute to priming the vole to form a new bond but are not sufficient to erase an existing bond as measured by partner preference. In addition, we demonstrate the utility of vTRAP for interrogating neuron-specific responses to bonding and separation that are obscured when analyzing bulk sequencing results. Together, these results expand the utility of monogamous prairie voles for future behavioral and molecular-genetic studies of partner loss and loss adaptation.

### Separation disrupts the continued strengthening of pair bonds but does not diminish baseline partner preference

We observed overall similarity and persistence in social behaviors exhibited by opposite-sex and same-sex paired male prairie voles where both types displayed a partner preference despite prolonged separation. The partner preference test, as employed in our study, may be insufficient to capture the uniquely rewarding aspect of pair bonds compared with other affiliative relationships such as sibling relationships (*Goodwin et al., 2019*). In part, this may reflect the coarse but commonly employed metric of proximity to the partner used to measure preference, without analyzing the valence or content of specific social interactions. A more nuanced assessment that includes naturalistic behaviors and/or captures the selectively motivating aspects of a pair bond would further elucidate differences in partner-directed behavior as a function of separation (*Brusman et al., 2021*; *Goodwin et al., 2019*; *Beery et al., 2021*; *Vahaba et al., 2022*).

However, the metrics employed here revealed that opposite-sex animals that remained paired exhibited increased partner huddle duration between baseline and long-term timepoints while the separated animals do not. Thus, while partner separation does not erase an established bond, it does disrupt the observed increase in partner huddle duration of remain paired males.

Prior work demonstrates the dissolution of partner preference in male voles after four weeks of separation (*Sun et al., 2014*). Yet the apparent contradiction we observed in our experiments may be due to differences in study design. In particular, we paired animals for two weeks before separation while prior studies had much shorter periods of cohabitation pre-separation (*Carter et al., 1995*; *Sun et al., 2014*). As demonstrated here and in other studies, pair bonds strengthen and mature over extended cohabitation periods, which may contribute to the observed perseverance of partner preference for weeks after separation (*Scribner et al., 2020*; *Brusman et al., 2021*). In addition, our females were tubally ligated in order to prevent pregnancy without disrupting other hormonal signaling. There is no consensus regarding the effects of pregnancy on male pair bonding behaviors (*Aragona and Wang, 2004*; *Brusman et al., 2021*; *DeVries et al., 1997*). However, partner preference in males is not affected by female ovariectomy (*Curtis, 2010*). Regardless, our results suggest that male voles remember and express selective affiliation despite prolonged isolation, though partner separation impedes the ongoing bond strengthening.

## Pair bonding results in consistent glia-associated transcriptional changes that erode following prolonged separation

Our studies begin to interrogate how dramatic shifts in socio-emotional context alter transcription in a mesolimbic hub—the NAc. We first asked whether bonding-induced gene expression is stable as long as pairs remain cohoused (*Resendez et al., 2016*; *Resendez et al., 2012*; *Aragona et al., 2006*; *Aragona et al., 2003*). Using three orthogonal analytical methods, we found remarkable consistency in the transcriptional pattern of opposite- vs same-sex pairs co-housed for 2 or 6 weeks. Prior work on social dominance suggests that such stable transcriptional changes are critical to maintain different behavioral states (*Cardoso et al., 2015*; *Zayed and Robinson, 2012*; *Tripp et al., 2018*). However, to our knowledge, this is the first demonstration that a similar conceptual underpinning may maintain mature pair bonds.

Consistently upregulated transcripts in pair bonded males are associated with glia and extracellular matrix organization while downregulated transcripts are associated with nucleoside processes, synaptic plasticity, and vesicle trafficking. Together, these terms implicate processes central to neuronal remodeling and synaptic plasticity. Of note, while our analyses do identify a number of mitochondrial GO terms, these terms are not as strongly implicated in our analyses as in previous reports (*Duclot et al., 2020*). Despite this, our study confirms a potential role for mitochondrial dynamics in the NAc in male prairie vole pair bonding. This is consistent with a model in which pair bonding and separation induce neuronal plasticity and synapse re-organization, which require large amounts of energy and induce mitochondrial redistribution within neurons (*Ülgen et al., 2023*).

We next found that pair bond transcriptional signatures erode as a function of separation duration. RRHO analyses demonstrated that there is considerable reduction in concordant pair bond gene expression following long-term but not short-term separation. When we asked which transcripts and associated biological processes erode over this timeframe, we found that they largely corresponded with those evident in the pair bond transcriptional signature. Specifically, transcripts associated with glia, extracellular matrix organization, synapse organization, and vesicle docking are stably expressed in pair bonded males but eroded following long-term separation. The erosion of the pair bond transcriptional signature over time in opposite-sex separated males may be central to adapting to partner loss in order to form a new pair bond if given the opportunity. It remains unclear though, how these changes may reflect or alter the component processes of pair bonding, such as partner memory, reward association, and/or motivation.

Our analyses strongly indicate a role for glia in regulating pair bonding and the response to partner separation. Transcripts associated with gliogenesis, glial cell differentiation, and myelination are upregulated in stable pair bonds, potentially reflecting the importance of glial cells in sculpting and refining neural circuits in response to social experience (*Loth and Donaldson, 2021*; *Kopec et al., 2018*; *Pohl et al., 2021*). Similar GO terms—with the addition of oligodendrocyte differentiation—are associated with eroded pair bond transcripts after long-term partner separation. Hypomyelination and oligodendrocyte dysfunction are seen in mice and rats after chronic social defeat or chronic social isolation and in individuals with depression (*Bonnefil et al., 2019*; *Lehmann et al., 2017*; *Zhou et al., 2021*). This appears reversible as social reintroduction rescued hypomyelination in mice (*Liu et al., 2012*). One mechanism for social deprivation-mediated hypomyelination and impaired oligodendrocyte maturation is by reduced flux through Erbb3-based signaling pathways (*Liu et al., 2012*; *Makinodan et al., 2012*). Of note, Erbb3 is one of 24 genes whose differential expression is conserved in taxonomically diverse monogamous species when compared to closely related non-monogamous relatives (*Young et al., 2019*). We find Erbb3 upregulated during pair bonding through short-term separation but it is no longer upregulated following long-term separation. Further, IPA analysis indicates that Erbb3 is an upstream regulator of the eroded genes. Thus, it is possible that long-term separation is downregulating Erbb3-associated pathways resulting in reduced myelination and disrupting oligodendrocytes specifically in opposite-sex-separated animals. An important future direction is whether bonding with a new partner re-engages these glial processes.

## Neurons have distinct transcriptional responses to bonding and loss

While there is an emerging appreciation for the role of glia in mediating behavior (*Thornton and Hughes, 2020*; *Bacmeister et al., 2022*; *Auguste et al., 2022*; *Hughes et al., 2018*; *Duncan and Emery, 2023*; *Kim et al., 2023*), these effects are largely due to modulation of neuronal activity,

and we reasoned that the prominence of glial-related terms may have masked important neuronal transcriptional changes in our dataset. To home in on neuronal transcriptional changes specifically, we developed vole-optimized translating ribosome affinity purification in prairie voles (vTRAP). We used DESeq, WGCNA, and a publicly available snRNAseq dataset to identify neuronal enriched genes, which show a 1.2–8.8 fold enrichment in neuronal pulldowns compared to the tissue-level sequencing data. We then queried expression changes specifically among these genes as a function of pair bonding and partner separation. With this enhanced sensitivity, we were able to identify three neuronal gene expression patterns that are differentially sensitive to partner loss. Notably, dopamine associated genes in cluster #2 are robustly upregulated in neurons after short-term partner loss, possibly reflecting a withdrawal-like state (*Potretzke and Ryabinin, 2019*; *Young et al., 2008*; *Lieberwirth and Wang, 2016*). This finding also underscores the discovery value of vTRAP because GO terms such as 'response to amphetamine', which indicates changes to dopaminergic genes, was a top 10 result, while the tissue level gene cluster predicts this GO term as the 34th result. Genes associated with anxiety and energy production in cluster #3 are upregulated only after long-term partner loss, possibly to facilitate seeking a new partner. Thus, together with our tissue level sequencing implicating the role of glia, vTRAP provides a more nuanced understanding of how specifically the pair bond neuronal transcriptional signature responds to partner separation over time.

## Limitations and future directions

Pair bonding and loss are complex and multifaceted processes, engaging diverse neural circuits in temporal- and context-specific ways. Yet transcriptomics provides a single snapshot of gene expression at one point in time in one context. So while tissue-level transcriptional analysis is an excellent tool to capture an averaged biological state, one limitation of our approach is that it is not ideal for examining more temporally-dynamic processes, such as the rapid information-processing that occurs during social decision making.

A notable aspect of our results is that the substantial erosion of pair bond transcription in the NAc after long-term separation is not matched by a similar reduction in partner-directed affiliative behaviors, despite evidence that changes in gene expression in this brain region help maintain bonds over time (*Aragona and Wang, 2004*; *Aragona et al., 2006*; *Aragona et al., 2003*). This apparent mismatch between behavior and transcription has a few possible biological explanations. First, while our focus on the NAc is well justified, other brain regions are also important for bonding (*Lim and Young, 2004*; *Tripp et al., 2021*; *Bosch et al., 2009*; *Gobrogge et al., 2009*). Specifically, the ventral tegmental area, medial pre-frontal cortex, and the hippocampus innervate the NAc and have known roles in social behavior. Transcriptional dynamics may differ within these regions, potentially contributing to bond persistence (*Okuyama et al., 2016*; *Smith et al., 2016*; *Kogan et al., 2000*; *Amadei et al., 2017*). In addition, pair bond transcription may induce other long-term biological changes that erode more slowly. These include long-lasting epigenetic memory as well as proteomic and/or structural changes that may contribute to behavioral persistence (*Tuesta and Zhang, 2014*; *Nestler et al., 2016*). Alternatively, transcriptional changes in a key subset of cells, which may be undetectable in tissue-level RNAseq, could drive bond persistence. More refined approaches, such as single-nucleus RNAseq, could greatly enhance our understanding of the cell types regulating complex behavioral persistence. All three of these hypothesized mechanisms for bond persistence represent important future lines of inquiry.

Previous work from our lab has shown that, when given the opportunity, male voles can form a new, stable pair bond that supplants the original bond if first isolated from their previous partner for at least four weeks (*Harbert et al., 2020*). However, in the current study design, male voles did not have an opportunity to find a new partner or form a new bond. Under such conditions, it may be advantageous to maintain behaviors that would enable the male to resume a bond if reunited with an absent partner. Speculatively, this framework optimizes reproductive opportunities by dually enabling a new bond to form *or* resuming an existing bond. It is likewise tempting to extrapolate this result into the context of human experience. We do not forget our previous bonds or their emotional valence; rather we integrate the loss in order to continue on with life.

Finally, oxytocin signaling in the NAc is required for pair bond formation and maintenance in male prairie voles (*Johnson et al., 2016*). While nonapeptide systems were not the focus of the current study, an intriguing follow-up question is how much of the pair bond transcription and its maintenance

is attributable to nonapeptide dynamics. Recent publication of pair bond formation in *Oxtr* null voles provides a unique opportunity to identify oxytocin-dependent and independent transcriptional changes that support pair bonds (*Berendzen and Manoli, 2022*).

## Conclusions

In sum, we provide additional insights into the neurogenomic processes that promote and maintain pair bonds and leverage these findings to gain a unique understanding of how the brain adapts to loss. We show that, in the absence of new bonding opportunities, the original pair bond will persist for at least 4 weeks post-partner-separation. However, prior work indicates that there is still an adaptive process occurring within this time frame that primes a vole to form a new bond if available (*Harbert et al., 2020*). Our work suggests that this adaptation is mediated at least in part by an erosion of bond-related transcription in mesolimbic systems that largely precedes bond dissolution at the behavioral level. Together, this work and future studies will help shed light on the pain associated with loss, as well as how we integrate grief in order to meaningfully reengage with life.

# Materials and methods

## Animals

Sexually naive adult prairie voles (Microtus ochrogaster, N=219: 161 M, 58 F, P60 - P168 at experiment start) were bred in-house in a colony originating from a cross between voles obtained from colonies at Emory University and University of California Davis, both of which were established from wild animals collected in Illinois. Animals were weaned at 21 days and housed in same-sex groups of two to four animals in standard static rodent cages (7.5 x 11.75 x 5 in.) with ad libitum water, rabbit chow (5326–3 by PMI Lab Diet) supplemented with alfalfa cubes, sunflower seeds, cotton nestlets, and igloos for enrichment until initiation of the experiment. All voles were aged between post-natal day 60 and ~180 at the start of the experiment. Throughout the experiment, animals were housed in smaller static rodent cages (11.0 x 8.0 x 6.5 in.) with ad libitum water, rabbit chow (5326–3 by PMI Lab Diet), and cotton nestlets. They were kept at 23–26°C with a 10:14 dark:light cycle to facilitate breeding. All procedures were performed in accordance with standard ethical guidelines (National Institutes of Health Guide for the Care and Use of Laboratory Animals) and approved by the Institutional Animal Care and Use Committee (IACUC) at the University of Colorado Boulder (protocol #2435).

## Tubal ligations

All females in opposite-sex pairs were tubally ligated to avoid confounds of pregnancy, but to keep the ovaries intact as to not impact hormonal function. Females were anaesthetized with a mixture of 2% Isoflurane and $O_2$ gas and depth of anesthetize was monitored by toe pinch throughout surgery. Prior to surgery, animals were weighed and given subcutaneous injections of the analgesic Meloxicam SR (4.0 mg/kg, Zoopharm) and saline (1 mL, Nurse Assist). Each fallopian tube was cauterized and bisected through lumbar inscitions on each side of the body using standard sterile practices. Animals were monitored once a day, for three days and staples were removed one week post-surgery. Females were given 2 weeks to recover before pairing with a male.

## Pairing

For the RNAseq portion of the experiment, opposite-sex, non-sibling pairs ranged in age between P98 and P168 (average ~P122) and same-sex, sibling pairs ranged in age between P60 to P163 (average ~P111) at pairing. For the behavioral experiments, all animals ranged in age between P85 and P168 (average ~121) at the time of pairing. For RNAseq and behavior testing, at pairing, pairs were transferred into smaller static rodent cages (11.0 in. x 8.0 in. x 6.5 in.) with ad libitum water and rabbit chow (5326–3 by PMI Lab Diet), a cotton nestlet, and an igloo. Female partners had been previously tubally ligated and were not induced prior to pairing. Partners cohabitated undisturbed, except for weekly cage changing performed by the same member of animal care staff, for 2 weeks prior to the baseline partner preference test (PPT). Same-sex pairings, sibling pairs from the same homecage were moved to smaller static rodent cages in the same way as the opposite-sex pairs. After the baseline PPT, RNAseq and behavior experiment animals that were either returned to their partner

(Remain Paired condition) or were separated into individual cages with no opportunity to see or smell their partner (Separated conditions).

## Separation for behavioral tests

Immediately following the baseline PPT, each pair was separated into fresh small static rodent cages (11.0 in. x 8.0 in. x 6.5 in.) with ad libitum water and rabbit chow (5326–3 by PMI Lab Diet), a cotton nestlet, and an igloo.

Separated animals were placed in an adjacent vole vivarium to eliminate visual and olfactory cues of the partner during the separation period. Each animal in the pair remained in their individual cages undisturbed for either 48 hours (short-term separation) or 4 weeks (long-term separation) except for weekly cage changes done by the same animal care staff member. At each separation time point, experimental animals performed a partner preference test (PPT) starting between 9:00 and 10:00 AM each day.

## Partner preference test (PPT)

PPT was carried out as described by Ahern et al. to assess selective partner affiliation prior to partner separation and were performed between the hours of 9:00 and 10:00 AM each testing day (*Ahern et al., 2009*). Each 3 hr test was recorded by overhead cameras (Panasonic WVCP304) that film two boxes simultaneously. The test animal is placed in the middle chamber for 10 minutes with the cage dividers still in place. The test begins when the cage dividers are removed and the test animal is allowed free range of all three chambers for the duration of the test. The movement of the test animal was tracked post-hoc using Topscan High-Throughput software (v3.0, Cleversys Inc) and frame by frame behavioral data was analyzed via a custom Python based script (https://github.com/donald-sonlab/separationRNAseq; copy archived at *Sadino, 2023*) to calculate the time spent huddling with each tethered animal. This data was then used to generate a number of behavior metrics including the partner preference score (Partner Huddle/Partner +Novel Huddle) (https://Figureshare.com/authors/zoe_donaldson_colorado_edu_Donaldson/6883910).

## Statistical analysis for behavioral experiments

All statistical analyses were performed in R (v4.0.4) using the base t.test package, base aov package, and the survival package (v3.2–7). Visualization was done using ggplot2 (v3.3.3), ggpubr (v0.4.0), and Adobe Illustrator (25.2.3). All statistical results can be found in *Supplementary file 1*. For PPT, we first determined if partner preference significantly changed from 50% at each time point (baseline, ST, or LT) and for each pairing type (OS or SS) by using an unpaired, one-sided Student's t-test against a null value of 50. We also performed a 2-way repeated measures ANOVA with Tukey's post-hoc test to determine main effects of, and interactions between, the pairing type and separation duration. Next, we calculated partner preference change scores for all conditions and compared the variance between pairing types using a Levene's test—the non-parametric equivalent of a F-test of variance. Finally, we determined changes to partner and novel huddling latency using the built in logRank test of the survminer package (v0.4.8).

## Generation of vole optimized hSyn-DIO-eGFP-RPL10a construct for vTRAP

Prior to generating the vTRAP viral construct, the nucleotide and amino acid sequences for RPL10a in mice and prairie voles were compared. To assess nucleotide homology, the mouse RPL10a cDNA sequence was aligned to the vole RPL10a cDNA sequence using BLASTN (v2.8.033) with a percent identity of 93% and an E-value of 0.0 (*Figure 4—figure supplement 1A*). The mouse RPL10a cDNA sequence was input into BLASTX (v2.8.033), and the resulting protein alignment against the prairie vole RPL10a amino acid sequence was 100% identical with an E-value of 3E$^{-145}$ (*Figure 4—figure supplement 1B*). Although the amino acid sequences are identical between mice and prairie voles, we generated the vTRAP construct using the prairie vole RPL10a cDNA sequence to account for possible species-specific differences in aminoacyl-tRNA concentrations.

To generate the TRAP construct, the DNA fragment containing the vole RPL10a sequence was inserted into the pAAV-hSyn-DIO-eGFP (Addgene plasmid #50457) backbone. The RPL10a fragment was designed for cloning using the NEBuilder HiFi Assembly method and consisted of a 5'

homology arm sequence, a linker sequence coding for SGRTQISSSSFEF, the vole RPL10a sequence (GenBank XM_005360358.1), and a 3' homology arm sequence (*Figure 4—figure supplement 1*). pAAV-hSyn-DIO-eGFP was digested using restriction endonucleases Asc1 (NEB) and BsrG1-HF (NEB), and the linearized plasmid was purified using the Zymogen Gel DNA Recovery Kit. The cloned plasmid, hereafter referred to as pAAV-hSyn-DIO-eGFPpvRPL10a (vTRAP),was generated using the NEBuilder HiFi DNA Assembly Cloning kit and subsequently transfected into 5-alpha F'Iq Competent *E. coli* (NEB). Transfected bacteria were selected for using carbenicillin (100 μg/ml) and harvested using the EZDNA Plasmid Mini Kit (Omega) or HiSpeed Plasmid Maxi Kit (Qiagen). The sequence of vTRAP was verified using Sanger sequencing (QuintaraBio) and restriction digest before being sent for viral packaging in an AAV1 serotype (Stanford Neuroscience Gene Vector and Virus Core).

## Viral injections

For stereotaxic surgery, adult prairie voles were anesthetized using isoflurane (3% induction, 1–2.5% maintenance) and depth of anesthesia was monitored by breathing and toe pinch reflex. After induction, animals were placed on a heating pad and given an analgesic (2 mg/kg Meloxicam SR, subcutaneous, Zoopharm) and sterile saline (1 mL, subcutaneous). Standard sterile surgical procedures were followed to expose and drill through the skull. Nucleus accumbens injection coordinates were adapted from the Allen Mouse Brain Atlas and validated prior to the experiment (AP: –1.7, ML:±1.0, DV: –4.7,–4.6, –4.5,–4.4, –4.3). For each DV coordinate, 100 nL of virus was dispensed at 0.1 nL/s for a total of 500 nL using a Nanoject II (Drummond Scientific). After the last injection, the needle remained in place for 10 minutes to allow for diffusion. The glass needle was then removed and the skin was sutured closed (vicryl sutures, eSutures), coasted with lidocaine and antibacterial ointment, and animals recovered in a cage on a heating pad with continued monitoring. The animal's health was monitored for 3 days following surgery and they were allowed to recover in their homecage for 2 weeks prior to the experiment start.

## Verification of TRAP expression

Cre-dependent expression of the vole optimized eGFP-RPL10a custom virus was validated in adult male prairie voles (n=3) using a within animal design. The stereotaxic injection coordinates for the NAc were adapted from the Paxinios and Franklin mouse brain atlas (the compact third edition) and were:+1.7 AP,+/-1 ML, –4.7 to –4.4 DV. The left NAc was injected with a viral cocktail of AAV1-DIO-eGFP-RPL10a (2 μL; 5.55E11 vg/mL) and AAV1-hSyn-iCre (2 μL; 3.9E12 vg/mL) while the right NAc only received the AAV1-DIO-eGFP-RPL10a virus (600 nL; 2.22E+12 vg/mL). Following 2 weeks of recovery to allow for robust viral expression the animal was perfused. The animal received a lethal injection of 0.2 mL 1:2 ketamine/xylazine and was perfused with 1 X PBS followed by 4% paraformaldehyde (in 1 X PBS, Electron Microscopy Sciences) and kept in 4% PFA overnight. The brain was then placed in 5 mL of 30% sucrose for 3 days prior to sectioning on a microtome (Leica JungSM2000R, 50 microns/slice). Sections were wet mounted onto Superfrost Plus glass slides (Thermo Fisher Scientific), cover slipped (ProLong Gold; Invitrogen), and allowed to dry overnight prior to imaging. Prepared slides were then sealed with nail polish (Electron Microscopy Sciences) and imaged using a Yokogawa CV1000 Confocal Scanner System (University of Colorado Light Microscopy Core Facility). Stitched images were produced using ImageJ (v1.51).

## Separation for RNAseq experiments

Immediately following the baseline PPT, each pair was moved to fresh small static rodent cages (11.0 in. x 8.0 in. x 6.5 in.) with ad libitum water and rabbit chow (5326–3 by PMI Lab Diet), a cotton nestlet, and an igloo. Remain Paired animals were kept with their partner while Separated animals were separated from their partner and moved to a fresh cage while their partner was immediately sacrificed to remove confounds of visual or olfactory cues. All animals remained in their individual cages undisturbed for either 48 hours (short-term separation) or 4 weeks (long-term separation) except for weekly cage changes done by the experimenter. After the appropriate separation time point, experimental animals were removed directly from their home cage for rapid decapitation and tissue harvesting.

## RNA preparation and sequencing

Tissue lysis and RNA isolation was adapted from Heiman and performed under RNase free conditions (*Heiman et al., 2014*). On each tissue collection day, 2 animals from each experimental cohort were processed, with the exception of same-sex 4-week paired animals who were all collected on the same day. Animals were removed from home cages, rapid decapped, and both hemispheres of the nucleus accumbens were hand dissected using sterile, RNase free razor blades and pooled. The NAc coordinates for dissections were:+1.7 AP,+/-1 ML, –4.7 to –4.4 DV with the TRAP dissections additionally guided by GFP fluorescence visualized with a fluorescent dissecting scope. During dissection, tissue was kept wet with dissection buffer (HEPES KOH, pH 7.3 20 mM, 1 X HBSS, Glucose 35 mM, NaHCO3 4 mM, CHX 10 uL of 1000 X stock, DEPC-treated water up to 10 mL). Tissue was added to 1 mL of ice-cold lysis buffer (DEPC-treated water, HEPES 20 mM, KCl 150 mM, MgCl2 (10 mM), DTT 5 μL of 1 M stock, CHX 10 μL of 1000 X stock, 1 protease inhibitor, RNasin 400 Units, and Superasin 800 Units), immediately homogenized using a Scilogex homogenizer, and kept on wet ice while the remaining animals for the day were processed. The homogenizer tip was thoroughly cleaned in between animals with DEPC-treated water and 70% ethanol to eliminate cross contamination between samples. Homogenized samples were centrifuged at 4 °C, for 10 min at ~3000 Xg and supernatant was transferred to new, pre-chilled tubes. 10% NP-40 (AG Scientific: #P-1505) and 300 mM DHPC (Avanti Polar Lipids, Inc: 850306 P-200 mg) were added to the supernatant and incubated on ice (4 °C) for 5 min. Samples were then centrifuged at 4 °C for 10 min at ~16,000 Xg and supernatant was transferred to new, pre-chilled tubes. RNA was then extracted and cleaned from the lysate supernatant using the Norgen Total RNA micro kit (cat. #) according to the manufacturer's instructions. RNA integrity was measured using an Agilent Tapestation prior to library preparation. RINs for all samples were >7.3 (average 8.8).

For TRAP samples, dissections and Input fraction RNA were prepared as above with the addition of GFP-guided dissections using a fluorescent dissecting scope. To perform immunoprecipitation pulldowns, affinity matrices were prepared fresh day-of using 300 μL of 10 mg/mL Streptavidin MyOne T1 Dynabeads (Invitrogen), 150 μL of 1 μg/μL Biotinylated Protein L (Fisher Scientific), and 50 μg each of eGFP antibodies Htz-19C8 and Htz-19F7 (Memorial-Sloan Kettering Monoclonal Antibody Facility) 2 hr prior to adding Input RNA sample. A small sample of Input RNA was saved for next day clean up (~250 μL/sample) and frozen at –80 °C overnight. The remaining Input RNA (~1 mL) was added to the GFP-conjugated beads and rotated overnight (16–18 hr) at 4 °C. The next day, the flowthrough was saved and stored at –80 °C and bound mRNA was released from the beads using a series of high salt buffer washes (4 washes; 20 mM HEPES, 150 mM KCl, 10 mM MgCl2, 10% NP-40, RNase-free water, protease inhibitor tablets, 0.5 mM DTT, 100 μg/μl cycloheximide). The beads were then incubated with the Norgen Total RNA micro kit SKP buffer (300 μL) for 10 min at room temperature. The Pull-down supernatant was then saved and processed as above using the kit manufacturer's instructions.

Library preparation for sequencing was done using the KAPA mRNA HyperPrep kit with polyA enrichment according to the manufacturer's instructions for a 100–200 bp insert size and using 1.5 μM TruSeq3-SE adapters. 71 samples were pooled and 3 runs of NextSeq V2 high output 75 cycle (1X75) sequencing produced a total read depth of ~20 M reads/sample. The Illumina sequencing was performed at the BioFrontiers Institute Next-Gen Sequencing Core Facility of University of Colorado Boulder. RNA quantity, RIN, and RNA amount used for library preparation can be found in *Supplementary file 1*.

## Sequence mapping and counting

For quality control, FastQC (0.11.8) and Trimmomatic-Src (v0.38) were used to analyze read quality and to trim the TruSeq3-SE adapters from all samples (*Bolger et al., 2014*). Additionally, each sequencing run was correlated to each other to ensure that there was not a significant difference in run composition. Reads were aligned to the prairie vole genome (Microtus_ochrogaster.MicOch1.0.dna.toplevel, released Feb 2017; accession GCA_000317375) using HiSat2 (2.1.0) with default options and individual runs for each animal were merged into a single sample using SAMtools (1.9) (*Kim et al., 2015*; *Danecek et al., 2021*). Reads were then counted using HTseq (0.11.2) with default options using the annotated prairie vole genome (Microtus_ochrogaster.MicOch1.0.95.gtf, released Feb 2017; accession GCA_000317375) (*Anders et al., 2015*). Genes with less than 10 read counts were filtered from downstream analysis and, where necessary, were normalized using transcripts per million (TPM).

Where possible, all Ensembl IDs were converted to prairie vole gene IDs using BioMart. If no gene ID was found, the original Ensembl ID was retained. Only genes with greater than 10 counts were included in subsequent analyses (total genes = 21152, genes >10 counts = 12128).

## Data analysis and visualization

All sequencing analysis was performed using R (v4.0.4) (*R:The R project for statistical computing, 2022*). The packages used and their specific parameters are described in the relevant sections below. Plots were made using base plot, ggplot2 (v3.3.3), EnhancedVolcano (1.8.0), RRHO2 (v1.0), cluster-Profiler (v3.18.1), and Adobe Illustrator (25.2.3) (*Cahill et al., 2018*; *Yu et al., 2012*; *Wickham, 2016*; *Blighe, 2021*).

## DESeq2

Differential gene expression was analyzed with the R package DESeq2 (v1.30.1) following the standard workflow (*Love et al., 2014*). To determine differential gene expression between cohorts a DESeq dataset of all animals was made with the design ~ParentCode + AgeAtPairing + Cohort to correct for the effect of parentage and age at initial pairing. Other batch effects (processing day, processing order, RIN, etc.) did not have sufficient influence on the model to include. Each result was a pair-wise contrast between the appropriate cohorts with the opposite-sex condition as the experimental and the same-sex condition as the reference. Differential gene expression was defined as >0.30 and<−0.30 log2FoldChange and p-value <0.05. This log2FoldChange threshold was used for two reasons. (1) A threshold of +/-0.20 can be successfully validated by qPCR (*Walker et al., 2018*) but (2) we chose a slightly higher threshold as it allows for smaller gene lists that appear more biologically relevant by Gene Ontology analysis. Differential gene expression was visualized using EnhancedVolcano. To determine if there was statistically significant overlap between differentially expressed gene lists compared to all genes (n=12,128) we performed a Fisher's exact test using the SuperExactTest (v1.0.7) package (*Wang et al., 2015*). Additionally, we used Morpheus (https://software.broadinstitute.org/morpheus) heatmaps to compare the expression profiles of each DESeq comparison using scaled log2FoldChange values. For each heatmap, a reference dataset ordered the remaining datasets such that each column is the same gene and the color represents the scaled $\log_2$FoldChange in each DESeq comparison. In *Figure 2F* the short-term pair bond was set as the reference and in *Figure 3F, J and L* the combined pair bond was set as the reference. Gene ontology and pathway analysis of relevant gene lists were performed using the R package enrichGO with *Mus musculus* ontology terms (*Yu et al., 2012*). Representative terms with an adjusted p-value of <0.05 were hand selected. Results were visualized using base R plots modified in Illustrator. All DESeq result data frames can be found in *Supplementary file 2*.

## Rank-rank hypergeometric overlap (RRHO)

RRHO is a threshold free approach used to identify genome-wide expression patterns between two ranked gene lists. RRHO analysis was performed using the RRHO2 (v1.0) package with parameters of stepsize = 100 and boundary = 0.02. First, transcripts from each DESeq comparison are ranked by their p-value and effect size direction:

$$\text{rankValue} = -1^*\log_{10}(\text{pvalue}) * \text{sign}(\text{Log}_2\text{FoldChange})$$

This results in a ranked list where the most significantly upregulated genes are at the top of the list and the most significantly downregulated genes are at the bottom of the list with each gene appearing only once. Next, two ranked gene lists are compared to each other using a hypergeometric distribution to determine the significance of the number of overlapping genes within defined sample populations. Sample populations are independently defined for each list. Each population consists of all of the genes that are ranked higher than a specified threshold. Thresholds successively move down a list at a user specified step size. For computation convenience, here we define our step size as 100 genes. However, the smallest possible step size is 1 gene and the largest possible step size is the number of genes found in the smallest list. The hypergeometric p-values resulting from comparing the List 1 genes (all genes above threshold x) to List 2 (all genes above threshold y) then populate a square matrix. As a result, the furthest bottom-left position of the matrix is the p-value of the overlap between genes ranked above the 1st threshold in each list and the furthest top-right position is the

overlap between genes ranked above the last threshold in each list. As another example, the furthest bottom-right position is the p-value comparing all genes above the last threshold of List 1 to all genes above the 1st threshold of List 2. The final matrix of hypergeometric p-values is visualized as a heatmap where List 1 is along the x-axis and List 2 is along the y-axis and each point represents a p-value. The axes are arranged such that genes that are upregulated in both lists are in the bottom-left quadrant (quadrant UU) and genes that are downregulated in both lists are in the top-right quadrant (quadrant DD). Genes with opposite regulation in each condition, up in one list but down in the other, are along the opposite diagonal (quadrants UD and DU) (*Figure 2E*). To visualize RRHO our heatmaps with a consistent p-value scale we set the scale maximum = 300 and the minimum = 1.

The genes in each quadrant were extracted to form UU, DD, UD, and DU gene lists for further analysis. Using these gene lists, the significant overlap between quadrants were determined using the SuperExactTest package. Gene ontology and pathway analysis of these gene lists were performed using the R package enrichGO with *Mus musculus* ontology terms. Representative terms with an adjusted p-value of <0.05 were hand selected. For the short-term vs long-term separation RRHO only UU and DD quadrant genes were used to filter the DESeq2 results of the pair bond, short-term separation, and long-term separation. We then further filtered the pair bond data by a log2FoldChange <–0.30 for the UU genes and >0.30 for the DD genes.

Additional information for RRHOs are in Figure S8. Unscaled p-value RRHOs can be found in Figure S8A-C. Since the p-value of the hypergeometirc distribution is sensitive to the number of genes, RRHO also plots a log-Odds ratio for all comparisons. Log-Odds ratio plots can be found in Figure S8D-F where the maximum scale was changed from 8 to 6 on all plots to better visualize values. Finally, to ensure that the signal we see in the RRHOs is not due to chance, we shuffled the ranked order of List 1 in each RRHO (Figure S8G-I). The resulting lack of signal throughout the RRHOs support that the signal we see in our experimental RRHOs is not due to chance.

## Pathway and upstream regulator analysis with ingenuity pathway analysis (IPA)

Ingenuity pathway analysis (Qiagen, Germantown, MD), was used to determine potential biological pathways and upstream regulators of DEGs and co-regulated genes identified using RRHO. Core expression analysis was applied for each gene list followed by comparison analysis for related lists to help identify common pathways and regulators across groups. For both, outcomes were filtered to include only those that reached an activation z-score of +/-2 and a corrected p-value of <0.05 (corrected by Benjamini Hochberg). All predicted biological pathways and upstream regulators are presented in *Supplementary file 3*.

## Weighted correlation network analysis (WGCNA)

A weighted gene co-expression network analysis was conducted using the R package WGCNA (v1.70–3) (*Langfelder and Horvath, 2008*). WGCNA analysis was done using the Input and Pulldown sequencing data from only animals in the vTRAP cohorts (opposite-sex remain paired 4 weeks, opposite-sex separated 4 weeks, and same-sex separated 4 weeks). A signed co-expression network was constructed using a power of 9 and minimum module size of 30 genes according to the author's standard module-trait relationship workflow. No samples were designated as outliers using complete clustering. Gene modules were then correlated to either the trait Input or Pulldown to determine the gene clusters associated with each sample type. To determine which correlation direction (positive or negative) is associated with which sample type, the significantly positively or negatively correlated modules were grouped and analyzed by Enrichr for cell types (*Chen et al., 2013*; *Kuleshov et al., 2016*; *Xie et al., 2021*). The top term for the positively correlated modules was nucleus accumbens while the top term for the negatively correlated modules was spinal cord. Therefore, the positively correlated modules represent the Pulldown samples, and by extension, neuronally enriched genes.

## Neuronal enriched genes throughout separation

To robustly define neuronally enriched genes we found the intersection of the vTRAP Input vs Pulldown enriched DEGs (*Figure 4B*) and the genes from the significantly, positively correlated WGCNA modules (*Figure 4D*). The resulting intersection—2365 genes— is referred to as 'neuronal enriched genes' and includes almost all of the vTRAP enriched DEGs. To further validate the neuronal enriched

genes we analyzed this gene list using Enrichr for cell types. The top two terms are nucleus accumbens and dorsal striatum confirming the specificity of this gene list.

We next wanted to follow the expression of these neuronal enriched genes throughout separation. First, since we only had vTRAP samples from specific cohorts in our experiment, we ensured that neuronal gene expression can be inferred from bulk sequencing of the same genes. To validate this, we filtered the Input and Pulldown samples of the opposite-sex 4-week separated cohort and compared the expression of the neuronal enriched genes using Morpheus (*Figure 4—figure supplement 2D*). The strong similarity in expression of the neuronal enriched genes using two different collection methods validated that bulk sequencing expression can be used to infer the neuron specific expression of those same genes. Second, to compare the expression patterns of the neuronal enriched genes to the bulk sequencing data we filtered the bulk data for only the neuronal genes. We then clustered the unfiltered bulk sequencing data, from 3F, and the neuronal enriched genes only bulk sequencing data using one minus Pearson correlation with complete linkage. For each neuronal cluster, we then found the corresponding cluster in the unfiltered expression data with the correspondence links visualized by a Sankey plot using the R package networkD3 (version 0.4)(*Allaire et al., 2017*). Third, we identified three clusters in the neuronal enriched data that had differential expression based on separation duration. We analyzed each gene list using Gene Ontology as previously described and highlighted specific genes associated with top terms. Which clusters were spotlighted in *Figure 4F* was determined by a combination of transcriptional patterns due to separation, which genes are in the cluster, and the resulting GO terms for each cluster. We identified clusters that represented three interesting patterns: (1.) down only after long-term pair bond separation (cluster 1) (2.) up due to acute pair separation (cluster 2) and (3.) up only after long-term pair bond separation (cluster 3).

## Acknowledgements

We thank Robin Dowell, Mary Allen, Ryan Logan, Xiangning Xue, and Gracie Sapp for their assistance in designing the experiments and analysis. We also thank Cayla Jo Paulson and Jessica Abazaris of the animal care staff at the University of Colorado Boulder. We thank the rest of the Donaldson lab for their feedback and support and the voles for their sacrifice. For vTRAP images we used the Molecular, Cellular, and Developmental Biology Light Microscopy Core at the University of Colorado Boulder with James Orth's assistance. This work was supported by NIH awards DP2OD026143, R01MH125423, and funds from the Whitehall Foundation and the Dana Foundation (to ZRD) and NIH award T32 GM008759-17/18 (to JMS).

## Additional information

### Funding

| Funder | Grant reference number | Author |
| --- | --- | --- |
| National Institutes of Health | T32 GM008759-17/18 | Julie M Sadino |
| Dana Foundation | | Zoe R Donaldson |
| Whitehall Foundation | | Zoe R Donaldson |
| National Institutes of Health | DP2OD026143 | Zoe R Donaldson |
| National Institutes of Health | R01MH125423 | Zoe R Donaldson |

The funders had no role in study design, data collection and interpretation, or the decision to submit the work for publication.

### Author contributions

Julie M Sadino, Conceptualization, Data curation, Formal analysis, Supervision, Investigation, Visualization, Methodology, Writing - original draft, Project administration, Writing - review and editing;

Xander G Bradeen, Conor J Kelly, Investigation; Liza E Brusman, Data curation, Formal analysis, Visualization, Writing - review and editing; Deena M Walker, Data curation, Software, Formal analysis, Visualization, Writing - review and editing; Zoe R Donaldson, Conceptualization, Resources, Data curation, Software, Formal analysis, Supervision, Funding acquisition, Visualization, Methodology, Writing - original draft, Project administration, Writing - review and editing

## Author ORCIDs
Julie M Sadino ⬡ http://orcid.org/0000-0002-2330-6198
Conor J Kelly ⬡ http://orcid.org/0000-0003-2410-7892
Zoe R Donaldson ⬡ http://orcid.org/0000-0001-6699-7905

## Ethics
All procedures were performed in accordance with standard ethical guidelines (National Institutes of Health Guide for the Care and Use of Laboratory Animals) and approved by the Institutional Animal Care and Use Committee (IACUC) at the University of Colorado Boulder (protocol #2435).

## Decision letter and Author response
Decision letter https://doi.org/10.7554/eLife.80517.sa1
Author response https://doi.org/10.7554/eLife.80517.sa2

## Additional files

### Supplementary files
• Supplementary file 1. Comprehensive statistical analyses and individual animal information. 10.6084/m9.figshare.19911541. Statistical analyses for behavior and transcription experiments including applicable post-hocs and version numbers of analysis packages used. Also includes relevant metrics for all experimental animals including information about excluded animals.

• Supplementary file 2. DESeq, RRHO, and Fisher's Exact Test intersection gene lists. 10.6084/m9.figshare.19911577. Workbook includes all gene lists used in analysis. For DESeq, all results dataframes and differentially expressed genes are included with all output metrics. RRHO tabs include genes in each list used to generate heatmaps and overlapping genes for each quadrant as well as genes in the Fisher's Exact Test intersections. Descriptions of all column names, the associated figure, and versions of packages used are in the last three tabs.

• Supplementary file 3. Ingenuity Pathway Analysis data. 10.6084/m9.figshare.19911583. Comprehensive output from IPA analyses for both Canonical Pathways and Upstream Regulators. Includes all terms with those that are statistically significant highlighted in red.

• MDAR checklist

### Data availability
All sequencing data is available on GEO (GSE192661). All lab specific code is available on the Donaldson Lab GitHub (https://github.com/donaldsonlab/separationRNAseq; copy archived at *Sadino, 2023*) and all metadata including comprehensive statistical analyses, animal information, differential gene expression, gene sets, and IPA analyses are available on the Donaldson lab Figshare. The vole optimized DIO-eGFP-pvRPL10a vector will be made available on Addgene.

The following datasets were generated:

| Author(s) | Year | Dataset title | Dataset URL | Database and Identifier |
|---|---|---|---|---|
| Sadino JM | 2022 | Supplemental Data 1 | https://doi.org/10.6084/m9.figshare.19911541 | figshare, 10.6084/m9.figshare.19911541 |
| Sadino JM | 2022 | Supplemental Data 2 | https://doi.org/10.6084/m9.figshare.19911541 | figshare, 10.6084/m9.figshare.19911577 |

*Continued on next page*

*Continued*

| Author(s) | Year | Dataset title | Dataset URL | Database and Identifier |
|---|---|---|---|---|
| Sadino JM | 2022 | Supplemental Data 3 | https://doi.org/10.6084/m9.figshare.19911541 | figshare, 10.6084/m9.figshare.19911583 |
| Sadino JM | 2023 | RNAseq data from: Prolonged partner separation erodes nucleus accumbens transcriptional signatures of pair bonding in male prairie voles | https://www.ncbi.nlm.nih.gov/geo/query/acc.cgi?acc=GSE192661 | NCBI Gene Expression Omnibus, GSE192661 |

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
