## [Editor Report]

This study focuses on an important but understudied topic, the biological basis of pair bonding and associated stress when the bond is lost. The authors present convincing evidence that transcriptional changes associated with the formation of a pair bond in voles degrade during prolonged separation but do not precisely mirror behavioral responses. Gene expression changes were associated with glia, steroid hormone signaling, and myelination, among others, providing critical guideposts for future investigations. This work will be of interest to behavioral neuroscientists and/or those interested in social behavior.

---

## [Decision Letter]

**Decision letter after peer review:**

Thank you for submitting your article "Prolonged partner separation erodes nucleus accumbens transcriptional signatures of pair bonding in male prairie voles" for consideration by *eLife*. Your article has been reviewed by 3 peer reviewers, including Margaret M McCarthy as Reviewing Editor and Reviewer #1, and the evaluation has been overseen by Catherine Dulac as the Senior Editor.

Essential revisions:

1) The disconnect between the behavioral responses in terms of endurance of the pair-bond, and the transcriptomics was of concern to the Reviewers. It was noted that there are subtle but perhaps important differences in the separation protocol used for the behavioral studies versus the transcriptomics. This combined with the behavioral data here not being consistent with the published literature undermines the current conclusions. The Reviewers felt that repeating the behavior but with the precise protocol used for the transcriptomics would allow for more direct comparison and strengthen the conclusions.

2) The value added by the use of TRAP was questionable. This approach should either be further validated or more strongly justified if it is to remain in the manuscript.

3) Some conclusions were overly anthropomorphized and an extensive literature on the roles of oxytocin and vasopressin in pair-bonding in this species was neglected. Changes in wording and emphasis combined with some new material in the Discussion could easily address this.

*Reviewer #1 (Recommendations for the authors):*

1) Prior work has implicated both the oxytocin and vasopressin systems in pair bonding, albeit in complicated ways, but there was both no discussion of that here nor an opportunity to provide a positive control since the nucleus accumbens does not include these systems. Perhaps the authors could discuss this as among the shortcomings.

2) Presumably the authors plan future studies using viral vectors to either over-express or repress candidate genes for manipulating the endurance of pair bonds. It is very unfortunate that the behavior did not correspond as expected. The authors' explanations for why this might be are legitimate but it does leave one wondering what to do next and highlights the challenge of such large sequencing studies.

*Reviewer #2 (Recommendations for the authors):*

This study was clearly a ton of work, and the authors marshall an impressive amount of data to address an important and understudied question. The combination of bioinformatics + bench work to get methods going in an emerging model is impressive.

It could help make the article more accessible to a wider readership if the authors could explain a little more clearly what vTRAP tells us that RNAseq doesn't. Why do we care about neuronal vs non-neuronal processes?

Some of the language about the transcriptomic results implies within-individual change although that can't be possible given that sampling is terminal, e.g. "…stably maintained while animals remained paired but eroded…" (abstract).

Interpretation of the transcriptional differences: clearly the transcriptional differences are not causal to the behavior that happened in the past. The decoupling of behavior and transcription offers some insights – the behavior remains the same (still shows behavioral signs of bonding) but the gene expression diverges; authors infer it has to do with the ability to establish a new bond but it's likely to be more heterogeneous? What about recovery from the loss, for example? Doesn't it seem likely that some of the DEGs reflect the ability to establish a new bond while others are probably about other processes, e.g. recovery?

Top left page 2: provide refs to support statements like "considering that other stable behavioral shifts…are underwritten by changes in transcription"

Why did the authors focus on males rather than females? The experimental design would be even better if they were brothers.

The sample size is ok with n=15, 16 pairs, but it would be helpful if the authors could provide some explicit acknowledgement of power (Figure 1).

R column, top of page 3: "latency to query"?

L column, page 3: need to explain why expected animals to have a preference for same-sex partners sooner before introducing the hypothesis.

L column, page 5: How to interpret opposite sex males initiated tumble fighting more quickly?

Page 5: please clearly state n for each treatment /time point group in the RNAseq analysis; presumably it's 50% of the n=15, 16 used in behavioral experiments?

Page 5, R column: which variables are being correlated here? "we calculated Rho values between the observed short-term pair bond and the shuffled…"

Page 5: how many individuals were excluded because they did not have baseline partner preference >50%?

Page 7: need to know experimental design for sequencing earlier, L column "on 8 groups of animals" – please remind us n for each group.

The major comparison is between same vs opposite sex housed pairs; therefore differences between treatment groups could reflect pair bonding OR they could reflect something about the sex of the partner. What about an additional control? They don't just differ in whether or not pair-bonded, but also in whether or not pair-bonded/housed/interacted with a male or a female

How did the authors choose which male member of the same-sex pair was used as the focal?

Page 8 L column: could this be correlated with the interesting behavioral results at the pair/individual level, i.e. pairs that were more bonded showed more of an effect than pairs that were less bonded?

Page 8, L column "the combined pair bond signature remains intact after short-term separation…" implies within-individual sampling.

Page 8, R column: this is a creative way to analyze RNAseq data!

vTRAP: Page 12, L column: It's neat that the authors were able to filter out genes so they could focus on genes expressed in neurons, but it seems like a lot of trouble to do all this work with cre and viruses – did they really learn that much more?

Page 12, R column: "the strong similarity in expression validates that bulk sequencing…" right – so did you really need to go to all the trouble of vTRAP?

Discussion on page 13 focuses on usual suspects/candidate genes: so why go to all this trouble to do this genomic approach if what really interests you are genes we already know are important? why not just have started with those?

Page 13: text here emphasizes that vTRAP didn't teach us anything particularly new we couldn't have figured out from the bulk RNASeq data alone. Here the authors emphasize that it's included in this paper to illustrate the utility of the method and make it available to the wider community of vole researchers, which is great but…. Does it belong in this paper?

Page 13 L column: the direct comparison of opposite versus same-sex paired animals provides a control for the effects of social context (alone vs with a pair) but it introduces another confound, namely the sex of the partner.

Page 13, R column: "correlation across partner preference behaviors…" seems like there could also be an opportunity to look for correlations between gene expression and behavior.

Page 14, L column: "transcripts associated with glia…" so can the effects of the experience of pair bonding vs pair bonding itself be teased apart? The experience of pair bonding could influence subsequent behavior via e.g learning -- there could be a transcriptional signal of that. There could also be a transcription signature of remembering that particular partner – can you differentiate between these two?

Sometimes the language goes a little too far "parsimonious with the human experience" page 15.

Methods: Tell us a little more about the colonies and genetic variation within them; how inbred are they?

*Reviewer #3 (Recommendations for the authors):*

1) The methods state that prairie voles were treated differently in the behavioral and transcriptomics studies. This might be why the authors do not see a strong behavioral degradation of pair bond-related behavior after long-term separation but do see a strong transcriptional signature. This reviewer understands that re-doing the behavioral study may not be feasible, this flaw should be addressed in the manuscript as a possible reason why the behavioral and transcriptomics results are not consistent. Further, the conclusions should be edited to fit this limitation.

2) While RRHO is helpful to assess overall patterns of transcriptional signatures across datasets, its utility for determining the exact transcripts is limited. For instance, even though the signal in the up-up quadrant in Figure 3F is stronger than the signal in the down-down quadrant in Figure 3G, there are more overlapping transcripts identified for a down-down. This is because of how RRHO determines the overlapping transcripts for its Venn diagram feature (by taking the point where the p-value is most significant and taking the list to the outside corner of that quadrant). It might be more helpful to use the RRHO quadrant information as a guide and then use traditional p-value cutoffs to look at overlapping transcripts. This would also reduce the number of transcripts fitting the desired pattern for follow-up experiments. The authors do this in the supplemental information which would be more helpful than the RRHO-identified overlapping transcripts presented in the main text. I suggest swapping these two approaches.

3) For the neuronal cluster analysis in Figure 4, the cluster 1 signal appears to be fairly weak, both in terms of the clustering in the neuronal enriched gene heatmap and the associated pathways. Why was this cluster selected? Further, it would be helpful to the reader if the authors included a description of each neuronal cluster in the figure (e.g., up in pair bond and ST separation, down in LT separation).

4) What is the signal for RRHO of short vs. long-term separation? Did I miss this? I feel that this is an important comparison to show.

5) In the volcano plots, it would be helpful to indicate the number of transcripts that are up or down-regulated on the figures.

6) Please include the coordinates for the NAc dissections.

7) The TRAP expression was verified in only one animal, which is not sufficient.

---

## [Author Response]

Essential revisions:1) The disconnect between the behavioral responses in terms of endurance of the pair-bond, and the transcriptomics was of concern to the Reviewers. It was noted that there are subtle but perhaps important differences in the separation protocol used for the behavioral studies versus the transcriptomics. This combined with the behavioral data here not being consistent with the published literature undermines the current conclusions. The Reviewers felt that repeating the behavior but with the precise protocol used for the transcriptomics would allow for more direct comparison and strengthen the conclusions.2) The value added by the use of TRAP was questionable. This approach should either be further validated or more strongly justified if it is to remain in the manuscript.3) Some conclusions were overly anthropomorphized and an extensive literature on the roles of oxytocin and vasopressin in pair-bonding in this species was neglected. Changes in wording and emphasis combined with some new material in the Discussion could easily address this.

We thank the reviewers for their thoughtful and constructive comments which have helped us improve our manuscript. In our revised manuscript, have respond to three main weaknesses:

1. We addressed the inconsistency in the experimental design across the behavior and the transcription experiments by repeating the behavior with an experimental timeline that more exactly matches that of the animals used in transcriptional studies and in which separated animals were housed in separate colony rooms. Briefly, we replicated our initial results and found that same- and opposite-sex pairs display a partner preference even after prolonged partner separation. However, by including an additional control group that was never separated, we also found that although partner directed huddle increased in animals that remain paired from two to six weeks, this was not true for animals that were separated, suggesting that separation disrupts normal bond strengthening. Thus while the behavior seems surprisingly robust in light of the observed transcriptional changes, we are confident that our core finding – that voles can maintain a preference despite prolonged separation – is indeed a reliable finding.

2. We further validated and justified our use of TRAP and our focus on the NAc as the sole brain region of investigation as articulated in more detail in the manuscript and below in our response to reviewers.

3. We revised the language throughout the manuscript, especially in the discussion, to reduce anthropomorphizing our results and interpretations. Below we have provided responses to specific concerns articulated by each reviewer.

Reviewer #1 (Recommendations for the authors):1) Prior work has implicated both the oxytocin and vasopressin systems in pair bonding, albeit in complicated ways, but there was both no discussion of that here nor an opportunity to provide a positive control since the nucleus accumbens does not include these systems. Perhaps the authors could discuss this as among the shortcomings.

A primary goal of our approach was to expand work in prairie voles beyond the narrow focus on nonapeptide systems. Justifying our approach to use RNA sequencing for an unbiased investigation, it was recently shown that the oxytocin receptor is not genetically required for pair bonding thus suggesting OXT-independent mechanisms of attachment in voles (*20*). We agree that an interesting area of future research will be to determine the contribution of nonapeptides to the observed pair bond transcriptional signature, but it was not a primary goal of the current study.

2) Presumably the authors plan future studies using viral vectors to either over-express or repress candidate genes for manipulating the endurance of pair bonds. It is very unfortunate that the behavior did not correspond as expected. The authors' explanations for why this might be are legitimate but it does leave one wondering what to do next and highlights the challenge of such large sequencing studies.

The opportunity to examine large biological datasets provides both challenges and opportunities. It this case, we have gained valuable timing information regarding the stability of pair bond transcription and the erosion of transcriptional signatures and robustness of partner preference following loss. Yet we are studying a

complicated system where we need additional information before confidently choosing and manipulating targets. In addition by repeating our behavioral experiments by including animals that remain paired we now know that separation impedes the strengthening of bonds providing us with an opportunity to potentially manipulate bond maturation. Such work is currently underway in our lab but beyond the scope of the current study.

Reviewer #2 (Recommendations for the authors):This study was clearly a ton of work, and the authors marshall an impressive amount of data to address an important and understudied question. The combination of bioinformatics + bench work to get methods going in an emerging model is impressive.

Thank you

It could help make the article more accessible to a wider readership if the authors could explain a little more clearly what vTRAP tells us that RNAseq doesn't. Why do we care about neuronal vs non-neuronal processes?

We feel that inclusion of TRAP adds substantially to this manuscript and to our understanding of the neuromolecular underpinnings of bonding and loss in the NAc. The value of this experiment is twofold. As noted by Reviewer 3, “the TRAP approach in prairie voles is novel and will provide a great resource to the research community.” The prairie vole community has just developed its first transgenic Cre lines, which could be paired with vTRAP to query bond-associated gene expression changes exclusively in Cre-expressing neurons (12). Second, we noticed a puzzle in our tissue-level data. The majority of cells in the NAc are neurons (13, 14), and the vast majority of pair bonding studies of this region have focused on neuronal phenotypes, but our transcriptional signatures were linked to changes in glial populations. Ultimately, changes in glia are likely to act via their interactions with neurons, and vTRAP enables us to query specifically the neuronal transcriptional changes within our data. Supporting that this provides novel insights into our datasets, when we cluster transcripts based on their expression profiles following short and long-term separation, we predict different GO terms from the tissue level and neuronally-enriched gene sets. For instance, the GO terms resulting from cluster 2 for neuronal genes (Fig 4) includes “response to amphetamine” within the top 10 results, but the same cluster of genes from tissue level sequencing predicts this GO term as the 34th result.

Some of the language about the transcriptomic results implies within-individual change although that can't be possible given that sampling is terminal, e.g. "…stably maintained while animals remained paired but eroded…" (abstract).

Thank you. It was not our intent to imply within animal changes but rather to clarify potential transcriptional signatures that characterize bonding and loss where some language signifying continuity is unavoidable (e.g. you cannot experience loss without a bond to lose). We have updated the verbiage throughout to clarify that our measurements are not evidence of within-animal change.

Interpretation of the transcriptional differences: clearly the transcriptional differences are not causal to the behavior that happened in the past. The decoupling of behavior and transcription offers some insights – the behavior remains the same (still shows behavioral signs of bonding) but the gene expression diverges; authors infer it has to do with the ability to establish a new bond but it's likely to be more heterogeneous? What about recovery from the loss, for example? Doesn't it seem likely that some of the DEGs reflect the ability to establish a new bond while others are probably about other processes, e.g. recovery?

We agree that there appears to be a mismatch between the behavioral and transcriptional results. Because of this we repeated the behavior (please see public response to Reviewer 1) with separate cohorts of animals for each separation timepoint and the addition of a remain paired cohort that stayed with their partner throughout the six weeks of the experiment. When we compared opposite-sex paired males that remained with their partner to those that were separated from their partner we observed that separation impedes the continual strengthening of the pair bond. We interpreted this result to mean that upon separation the original pair bond is not lost per-say it is not being maintained in the way that it would be if the animals had stayed paired. Thus, the behavior now more closely resembles the transcriptional result of a residual pair bond signature that erodes over the course of separation. Future experiments will focus on a more nuanced understanding of how transcription can influence behavior to remain pliable in multiple social contexts. We addressed this comment in the discussion as follows:

“Pair bonding and loss are complex and multifaceted processes, engaging diverse neural circuits in temporal- and context-specific ways. Yet transcriptomics provides a single snapshot of gene expression at one point in time in one context. So while tissue-level transcriptional analysis is an excellent tool to capture an averaged biological state, one limitation of our approach is that it is not ideal for examining more temporally-dynamic processes, such as the rapid information-processing that occurs during social decision making.”

Top left page 2: provide refs to support statements like "considering that other stable behavioral shifts…are underwritten by changes in transcription"

Thank you for bringing this oversight to our attention. Three references have been added to this statement.

Why did the authors focus on males rather than females? The experimental design would be even better if they were brothers.

The methods state that the same-sex pairs were indeed siblings though we updated the language in the introduction to explicitly state that the same-sex pairs were siblings: “Here we map the trajectory of the pair bond transcriptional profile by comparing opposite-sex paired male to same-sex sibling pairs.” We plan to carry out similar studies in females, but colony size consideration and the complexity of the current dataset contributed to our decision to focus on males initially.

The sample size is ok with n=15, 16 pairs, but it would be helpful if the authors could provide some explicit acknowledgement of power (Figure 1).

We performed a power analysis prior to undertaking the study in order to estimate the group size needed to obtain significant partner preference at (α, Β blah blah). In general, it is not recommended to do posthoc power estimation. If a study was underpowered to detect the true effect size, a post-hoc power analysis of significant (p < 0.05) data is likely to subsequently overestimate power. Mathematically, the power at the observed effect is a function of the p-value. Instead, we have provided effect sizes for significant results, reported in the supplementary statistics table.

R column, top of page 3: "latency to query"?

We changed “query” to “determine” for clearer language.

L column, page 3: need to explain why expected animals to have a preference for same-sex partners sooner before introducing the hypothesis.

Thank you for pointing out our oversight in not including this information. We have added the following clause to the text:

“Similarly, we anticipated that same-sex sibling pairs would have a partner preference at baseline due to familiarity but would exhibit a faster and more robust loss of partner preference when separated.”

L column, page 5: How to interpret opposite sex males initiated tumble fighting more quickly?

We believed that repeating the partner preference test data was a crucial revision, and we correspondingly chose to avoid any potential confounds. Thus we did not to measure resident intruder due to concerns of that aggressive homecage interactions when the partner was removed could have unintended effects on affiliative behavior. All aggression data has been removed from the manuscript.

Page 5: please clearly state n for each treatment /time point group in the RNAseq analysis; presumably it's 50% of the n=15, 16 used in behavioral experiments?Page 5: how many individuals were excluded because they did not have baseline partner preference >50%?

Group size and excluded animals for all transcription experiments are now clearly stated in the results of the manuscript. Additionally, cohort numbers can be found in the partner preference graphs of Figures 2 and 3.

In addition to listing the excluded animals in Supplemental Data S1, we have added the numbers of excluded animals that had a baseline partner preference <50% in the text as well. For the combined pair bond analysis the verbiage now reads as:

“For consistency, we limited transcriptional assessment to voles that had a baseline partner preference >50% (Figure 2B; Table S1) (OS n = 15; SS n = 11. Excluded for PPT <50% OS n = 3; SS n = 8).” For the separation analysis the text now reads “As before, we only included animals with a baseline partner preference >50% (Figure 3B) (OS n = 12; SS n = 11. Excluded for PPT <50% OS n = 4; SS n = 5).”

Page 5, R column: which variables are being correlated here? "we calculated Rho values between the observed short-term pair bond and the shuffled…"

The variable used to calculate the Rho values was the log2FoldChange as calculated by DESeq2 during differential gene expression analysis. We have revised the sentence to include this information:

“We shuffled each animal’s cohort identity at the long-term timepoint and calculated Rho values of the log_2_FoldChange of differential gene expression between the observed short-term pair bond and the shuffled long-term pair bond over 1000 iterations”

Page 7: need to know experimental design for sequencing earlier, L column "on 8 groups of animals" – please remind us n for each group.

The experimental design is described in the first paragraph of the Results section and is graphically represented in Figures 2A and 3A. The verbiage in question was changed to “on all animals in this study (n = 49).”

The major comparison is between same vs opposite sex housed pairs; therefore differences between treatment groups could reflect pair bonding OR they could reflect something about the sex of the partner. What about an additional control? They don't just differ in whether or not pair-bonded, but also in whether or not pair-bonded/housed/interacted with a male or a femalePage 13 L column: the direct comparison of opposite versus same-sex paired animals provides a control for the effects of social context (alone vs with a pair) but it introduces another confound, namely the sex of the partner.

Please see our response to your request (point 1) in the public response.

How did the authors choose which male member of the same-sex pair was used as the focal?

The following sentence was added to methods under section RNA preparation and sequencing:

“The focal animal was chosen at random from each pair prior to the beginning of the experiment.”

Page 8 L column: could this be correlated with the interesting behavioral results at the pair/individual level, i.e. pairs that were more bonded showed more of an effect than pairs that were less bonded?Page 13, R column: "correlation across partner preference behaviors…" seems like there could also be an opportunity to look for correlations between gene expression and behavior.

Please see our public response point R2.1.

Page 8, L column "the combined pair bond signature remains intact after short-term separation…" implies within-individual sampling.

We have updated the verbiage to make it clearer that we are examining across groups and not within animals. The text now reads:

“Specifically, when genes are ordered based on their degree of down- or up-regulation in the pair bond group we found that gene expression was largely indistinguishable in short-term separated animals but the pair bond signature(Rho = 0.32, p < 2.2X10^-16^) is largely eroded in long-term separated animals (Rho = 0.048, p = 1.6X10^-7^) (Figure 3D).”

Page 8, R column: this is a creative way to analyze RNAseq data!

Thank you!

vTRAP: Page 12, L column: It's neat that the authors were able to filter out genes so they could focus on genes expressed in neurons, but it seems like a lot of trouble to do all this work with cre and viruses – did they really learn that much more?Page 12, R column: "the strong similarity in expression validates that bulk sequencing…" right – so did you really need to go to all the trouble of vTRAP?Page 13: text here emphasizes that vTRAP didn't teach us anything particularly new we couldn't have figured out from the bulk RNASeq data alone. Here the authors emphasize that it's included in this paper to illustrate the utility of the method and make it available to the wider community of vole researchers, which is great but…. Does it belong in this paper?

We feel strongly that the vTRAP data adds substantially although we agree that this could have been more clearly articulated that it was in the prior version of the manuscript. The following is copied verbatim from our public response and is now included in the Discussion section: We feel that inclusion of TRAP adds substantially to this manuscript and to our understanding of the neuromolecular underpinnings of bonding and loss in the NAc. The value of this experiment is twofold. As noted by Reviewer 3, “the TRAP approach in prairie voles is novel and will provide a great resource to the research community.” The prairie vole community has just developed its first transgenic Cre lines, which could be paired with vTRAP to query bond-associated gene expression changes exclusively in Cre-expressing neurons (*15*). Second, we noticed a puzzle in our tissue-level data. The majority of cells in the NAc are neurons (*16*, *17*), and the vast majority of pair bonding studies of this region have focused on neuronal phenotypes, but our transcriptional signatures were linked to changes in glial populations. Ultimately, changes in glia are likely to act via their interactions with neurons, and vTRAP enables us to query specifically the neuronal transcriptional changes within our data. Supporting that this provides novel insights into our datasets, when we cluster transcripts based on their expression profiles following short and long-term separation, we predict different GO terms from the tissue level and neuronally enriched gene sets. For instance, the GO terms resulting from cluster 2 for neuronal genes (Figure 4) includes “response to amphetamine” within the top 10 results, but the same cluster of genes from tissue level sequencing predicts this GO term as the 34th result.

Discussion on page 13 focuses on usual suspects/candidate genes: so why go to all this trouble to do this genomic approach if what really interests you are genes we already know are important? why not just have started with those?

We take a slightly different perspective. Our identification of such “usual suspects” demonstrates that our results are consistent with the known biology of pair bonding. This dually validates prior work and provides opportunities to expand our understanding by looking at other features of our dataset. In particular, our genomic approach has expanded our understanding via: (1) identification of transcriptional patterns important for regulating behavioral states, (2) identification of upstream regulators that modulate such multi-gene expression patterns, and (3) insight into potential biological processes not previously implicated in pair bonding, such as oligodendrogenesis/myelination, which would be unlikely to result from focusing on candidate genes alone.

Page 14, L column: "transcripts associated with glia…" so can the effects of the experience of pair bonding vs pair bonding itself be teased apart? The experience of pair bonding could influence subsequent behavior via e.g learning -- there could be a transcriptional signal of that. There could also be a transcription signature of remembering that particular partner – can you differentiate between these two?

We interpret this to comment to mean that (1) pair bond formation is distinct from bond maintenance, with the former likely reflecting complex learning processes and the latter evoking homeostatic mechanisms that reinforce the bond. And (2) bonding requires multiple component processes (social recognition memory, motivation, reward association, etc). Towards the former, we are currently investigating the hypothesis that gliogenesis/myelination may be required for bond formation/maturation, consistent with its role in other complex forms of learning and long-term memory (*21*–*23*). These results will be presented in a separate manuscript. Towards the latter, the behavioral test we employed (the partner preference test) is the gold standard to assessing pair bonds, but it cannot dissociate the different component processes that contribute to bonding. An important area of follow-up research will be to ask how transcriptional or other neurobiological changes contribute to the component processes and their integration in bonding. However, we feel that this question is beyond the scope of the current experiment.

Sometimes the language goes a little too far "parsimonious with the human experience" page 15.

We thank the reviewer for encouraging us to remain objective with our interpretations. We have addressed anthropomorphic wording in the manuscript with the following revision:

We changed “Such an explanation is parsimonious with the human experience.“ to

“Speculatively, this framework optimizes reproductive opportunities by dually enabling a new bond to form *or* resuming an existing bond*.* It is likewise tempting to extrapolate this result into the context of human experience. We do not forget our previous bonds or their emotional valence; rather we integrate the loss in order to continue on with life.”

Methods: Tell us a little more about the colonies and genetic variation within them; how inbred are they?

The current methods state that *“*Sexually naive adult prairie voles (Microtus ochrogaster, N = 219: 161M, 58F, P60 – P168 at experiment start) were bred in-house in a colony originating from a cross between voles obtained from colonies at Emory University and University of California Davis, both of which were established from wild animals collected in Illinois.*”*

We are careful to maintain an outbred colony. In particular, we maintain pedigrees, and we routinely breed in new, wild-derived voles when animals approach ~20% shared background among non-first-degree relations (approximately every 2 years). However, we have not performed genome sequencing to calculate their degree of relatedness.

Reviewer #3 (Recommendations for the authors):1) The methods state that prairie voles were treated differently in the behavioral and transcriptomics studies. This might be why the authors do not see a strong behavioral degradation of pair bond-related behavior after long-term separation but do see a strong transcriptional signature. This reviewer understands that re-doing the behavioral study may not be feasible, this flaw should be addressed in the manuscript as a possible reason why the behavioral and transcriptomics results are not consistent. Further, the conclusions should be edited to fit this limitation.

The following is modified from the public response: The experimental designs employed to assess behavior and transcription differed, which may have contributed to the apparent mismatch in our results when comparing these two levels of biology. Thus we repeated our behavioral assessment using a design that directly matches the one we used in our transcriptional assessment. Specifically, we included a set of “remain paired” controls, separated partners were kept in separate colony rooms to ensure no possible access to partner-associated sensory cues (visual, auditory, olfactory), and separate cohorts of animals were used for short- and long-term separation. This design avoided partner re-introduction during the short-term partner preference test. In short, we replicated our initial findings; prairie voles will display a partner preference for same- or opposite-sex partner even after prolonged separation. Of note, animals that remained paired showed an increase in partner-directed affiliation (huddle time), which was not observed in separated animals suggesting that separation impairs the normal strengthening of bonds with time. Results associated with Figure 1 and the relevant discussion points have been updated in the manuscript.

2) While RRHO is helpful to assess overall patterns of transcriptional signatures across datasets, its utility for determining the exact transcripts is limited. For instance, even though the signal in the up-up quadrant in Figure 3F is stronger than the signal in the down-down quadrant in Figure 3G, there are more overlapping transcripts identified for a down-down. This is because of how RRHO determines the overlapping transcripts for its Venn diagram feature (by taking the point where the p-value is most significant and taking the list to the outside corner of that quadrant). It might be more helpful to use the RRHO quadrant information as a guide and then use traditional p-value cutoffs to look at overlapping transcripts. This would also reduce the number of transcripts fitting the desired pattern for follow-up experiments. The authors do this in the supplemental information which would be more helpful than the RRHO-identified overlapping transcripts presented in the main text. I suggest swapping these two approaches.

Please see our public response point R3.2.

3) For the neuronal cluster analysis in Figure 4, the cluster 1 signal appears to be fairly weak, both in terms of the clustering in the neuronal enriched gene heatmap and the associated pathways. Why was this cluster selected? Further, it would be helpful to the reader if the authors included a description of each neuronal cluster in the figure (e.g., up in pair bond and ST separation, down in LT separation).

Which clusters were spotlighted in Figure 4F was determined by a combination of transcriptional patterns due to separation, which genes are in the cluster, and the resulting GO terms for each cluster. We identified clusters that represented three interesting patterns: (1.) down only after long-term pair bond separation (cluster 1) (2.) up due to acute pair separation (cluster 2) and (3.) up only after long-term pair bond separation (cluster 3). While other clusters could have been used, we felt that these were important transcriptional patterns to highlight in the context of the rest of our results. Further, we noted that cluster 1 contains *Shank3*, a gene that has been linked to autism and for which mice containing the null allele show social behavior deficits (*24*). We reasoned that the expression of *Shank3*, and other genes with a similar transcriptional pattern after partner separation, may be of interest to a broad audience. Finally, we ran GO analysis on each cluster. Unfortunately, at this last step, none of the GO terms for cluster 1 were significant though this is likely due to the cluster’s small size. This is indicated by the light shading, but we felt it was worth including for the sake of transparency.

We have added the following description of cluster selection taken from this response has been added to the “Neuronal enriched genes throughout separation” methods:

“Which clusters were spotlighted in Figure 4F was determined by a combination of transcriptional patterns due to separation, which genes are in the cluster, and the resulting GO terms for each cluster. We identified clusters that represented three interesting patterns: 1.) down only after long-term pair bond separation (cluster

1) 2.) up due to acute pair separation (cluster 2) and 3.) up only after long-term pair bond separation (cluster 3).”

The following description of Cluster 1 is provided in the text:

“Cluster 1 consists of genes that were upregulated in opposite-sex paired males during pair bonding and short-term separation and downregulated following longterm separation.”

The following description of Cluster 2 is provided in the text:

“Cluster 2 transcripts are downregulated in pair bonded males and then robustly upregulated after short-term partner loss and, to a lesser extent, at the longterm timepoint indicating an acute transcriptional response to pair bond disruption.”

The following description of Cluster 3 is provided in the text:

“Finally, Cluster 3 contains genes that were downregulated in pair bonded males and became upregulated only after long-term opposite-sex partner separation, representing transcripts that may help prime the vole to form a new bond.”

Additionally, we updated cluster labels in Figure 4 to be more descriptive.

4) What is the signal for RRHO of short vs. long-term separation? Did I miss this? I feel that this is an important comparison to show.

We thank the reviewer for pointing this out. While we had looked at this in some of our initial analyses, we ultimately chose to focus on the question of what happens to the pair bond signature upon short- or long-term separation, thus omitting this secondary analysis. Based on the above question, we revisited this RRHO, which is now provided in Figure 3 —figure supplement 5. Notably, we see substantial overlap in the UU and DD quadrants in the short-term to long-term separation comparison. This pattern is particularly enriched for overlapping DD transcripts, which is the inverse of what we see in the pair bond::short term RRHO. One likely explanation for this potential overlap, especially in the DD quadrant, is that we are picking up on genes that have already eroded by the short-term separation timepoint and remain eroded after long-term separation. Supporting this interpretation, we took the UU and DD quadrant gene lists (referred to as stable separation genes) and filtered the DESeq2 results of the pair bond, short-term separation, and long-term separation (the same process as Figure 3H and I) for these genes. We identified an interesting pattern in that the genes that were most highly up- or downregulated in the pair bond showed a reversal of expression after short-term separation, correspondingly falling in the opposite UU or DD quadrant post-separation. Thus we filtered the stable separation genes for each condition by a log2FoldChange < -0.30 in the UU quadrant and > 0.30 in the DD quadrant and assessed the transcriptional state of these genes in the pair bond. Figure 3 —figure supplement 5B and C show that the genes that were most downregulated in the UU quadrant show a sharp inversion of expression after short-term separation that is maintained through long-term separation and vice-versa for the DD quadrant genes. Thus, there are subsets of transcripts that are up- or downregulated in pair bonded animals that are already exhibiting erosion after short-term separation. We further asked which biological processes are represented by the genes in Figure 3 —figure supplement 5D and E to investigate which processes are sensitive specifically to short-term separation. We found that UU genes that are then downregulated are associated with dopaminergic processes and DD genes that are then upregulated are associated with neuronal rearrangement processes. Thus, this RRHO analysis enriches our interpretation of transcriptional erosion of pair bond signatures following partner separation. Also related to point R3.2, overall we observed only slight to moderate erosion at the short-term timepoint, which was masked when relying exclusively on a strict p-value cutoff as in Supplemental Figure 6F. Rather, our rigorous experimental design, which has been noted as a strength of this manuscript, allows for more interrogation of the temporal complexity of these transcriptional signatures via threshold-free approaches.

5) In the volcano plots, it would be helpful to indicate the number of transcripts that are up or down-regulated on the figures.

We have added the number of upregulated and downregulated genes to each volcano plot. Additionally, gene lists and the associated differential expression values can be found in Supplemental Data 2.

6) Please include the coordinates for the NAc dissections.

The section “RNA preparation and sequencing” now includes the following description: “The NAc coordinates for dissections were: +1.7 AP, +/- 1 ML, -4.7 to -4.4 DV with the TRAP dissections additionally guided by GFP fluorescence visualized with a fluorescent dissecting scope.”

7) The TRAP expression was verified in only one animal, which is not sufficient.

We interpret this to mean that there are concerns that we only confirmed the Cre-dependence of the TRAP construct in a single animal as all other aspects of the TRAP work involved significantly more animals. We note that most labs rarely confirm Cre-dependence of vectors in more than one or two animals as the results, including those shown in Figure 4 —figure supplemental 2A, are typically definitive (i.e. no expression in the absence of Cre, expression in the presence of Cre). In addition to the images shown in Figure 4 —figure supplemental 2A, we used fluorescent guided dissection to harvest tissue/mRNA, serving as an additional visual confirmation of RPL10-GFP expression in the animals used to generate Figure 4. Additionally, a collaborating lab also confirmed that this vector also expresses in rats when Cre-recombinase is present. Finally, we performed two additional surgeries to confirm that TRAP is only expressed in the presence of Crerecombinase since the prior submission.

References

1. J. L. Scribner, E. A. Vance, D. S. W. Protter, W. M. Sheeran, E. Saslow, R. T. Cameron, E. M. Klein, J. C. Jimenez, M. A. Kheirbek, Z. R. Donaldson, A neuronal signature for monogamous reunion. Proceedings of the National Academy of Sciences. 117, 11076–11084 (2020).

2. A. M. Borie, S. Agezo, P. Lunsford, A. J. Boender, J.-D. Guo, H. Zhu, G. J. Berman, L. J. Young, R. C. Liu, Social experience alters oxytocinergic modulation in the nucleus accumbens of female prairie voles. Curr Biol. 32, 1026-1037.e4 (2022).

3. in vivo imaging identifies temporal signature of D1 and D2 medium spiny neurons in cocaine reward | PNAS, (available at https://www.pnas.org/doi/10.1073/pnas.1521238113).

4. M.-F. O’Connor, D. K. Wellisch, A. L. Stanton, N. I. Eisenberger, M. R. Irwin, M. D. Lieberman, Craving love? Enduring grief activates brain’s reward center. NeuroImage. 42, 969–972 (2008).

5. T. Hikida, S. Yao, T. Macpherson, A. Fukakusa, M. Morita, H. Kimura, K. Hirai, T. Ando, H. Toyoshiba, A. Sawa, Nucleus accumbens pathways control cell-specific gene expression in the medial prefrontal cortex. Sci Rep. 10, 1838 (2020).

6. C. Baimel, L. M. McGarry, A. G. Carter, The Projection Targets of Medium Spiny Neurons Govern Cocaine-Evoked Synaptic Plasticity in the Nucleus Accumbens. Cell Reports. 28, 2256-2263.e3 (2019).

7. N. S. Lee, N. L. Goodwin, K. E. Freitas, A. K. Beery, Affiliation, aggression, and selectivity of peer relationships in meadow and prairie voles. Frontiers in Behavioral Neuroscience. 13 (2019), doi:10.3389/fnbeh.2019.00052.

8. O. J. Bosch, H. P. Nair, T. H. Ahern, I. D. Neumann, L. J. Young, The CRF System Mediates Increased Passive Stress-Coping Behavior Following the Loss of a Bonded Partner in a Monogamous Rodent. Neuropsychopharmacology. 34, 1406–1415 (2009).

9. O. J. Bosch, J. Dabrowska, M. E. Modi, Z. V. Johnson, A. C. Keebaugh, C. E. Barrett, T. H. Ahern, J. Guo, V. Grinevich, D. G. Rainnie, I. D. Neumann, L. J. Young, Oxytocin in the nucleus accumbens shell reverses CRFR2-evoked passive stress-coping after partner loss in monogamous male prairie voles. Psychoneuroendocrinology. 64, 66–78 (2016).

10. A. J. Grippo, B. S. Cushing, C. S. Carter, Depression-like behavior and stressor-induced neuroendocrine activation in female prairie voles exposed to chronic social isolation. Psychosomatic Medicine. 69, 149– 157 (2007).

11. A. J. Grippo, D. Gerena, J. Huang, N. Kumar, M. Shah, R. Ughreja, C. Sue Carter, Social isolation induces behavioral and neuroendocrine disturbances relevant to depression in female and male prairie voles. Psychoneuroendocrinology (2007), doi:10.1016/j.psyneuen.2007.07.004.

12. K. Horie, K. Inoue, S. Suzuki, S. Adachi, S. Yada, T. Hirayama, S. Hidema, L. J. Young, K. Nishimori, Oxytocin receptor knockout prairie voles generated by CRISPR/Cas9 editing show reduced preference for social novelty and exaggerated repetitive behaviors. Horm Behav. 111, 60–69 (2019).

13. K. E. Savell, J. J. Tuscher, M. E. Zipperly, C. G. Duke, R. A. Phillips, A. J. Bauman, S. Thukral, F. A. Sultan, N. A. Goska, L. Ianov, J. J. Day, A dopamine-induced gene expression signature regulates neuronal function and cocaine response. Science Advances. 6, eaba4221 (2020).

14. Single-Cell RNA-Seq Uncovers a Robust Transcriptional Response to Morphine by Glia: Cell Reports, (available at https://www.cell.com/cell-reports/fulltext/S2211-1247(18)31384-6?_returnURL=https%3A%2F%2Flinkinghub.elsevier.com%2Fretrieve%2Fpii%2FS2211124718313846%3 Fshowall%3Dtrue).

15. S. L. Fulton, S. Mitra, A. E. Lepack, J. A. Martin, A. F. Stewart, J. Converse, M. Hochstetler, D. M. Dietz, I. Maze, Histone H3 dopaminylation in ventral tegmental area underlies heroin-induced transcriptional and behavioral plasticity in male rats. Neuropsychopharmacology. 47, 1776 (2022).

16. S. G. Caradonna, T.-Y. Zhang, N. O’Toole, M.-J. Shen, H. Khalil, N. R. Einhorn, X. Wen, C. Parent, F. S. Lee, H. Akil, M. J. Meaney, B. S. McEwen, J. Marrocco, Genomic modules and intramodular network concordance in susceptible and resilient male mice across models of stress. Neuropsychopharmacol. 47, 987–999 (2022).

17. J. S. Wang, T. Kamath, C. M. Mazur, F. Mirzamohammadi, D. Rotter, H. Hojo, C. D. Castro, N. Tokavanich, R. Patel, N. Govea, T. Enishi, Y. Wu, J. da Silva Martins, M. Bruce, D. J. Brooks, M. L. Bouxsein, D. Tokarz, C. P. Lin, A. Abdul, E. Z. Macosko, M. Fiscaletti, C. F. Munns, P. Ryder, M. KostAlimova, P. Byrne, B. Cimini, M. Fujiwara, H. M. Kronenberg, M. N. Wein, Control of osteocyte dendrite formation by Sp7 and its target gene osteocrin. Nat Commun. 12, 6271 (2021).

18. D. A. Gallegos, M. Minto, F. Liu, M. F. Hazlett, S. Aryana Yousefzadeh, L. C. Bartelt, A. E. West, Cell-type specific transcriptional adaptations of nucleus accumbens interneurons to amphetamine. Mol Psychiatry, 1–15 (2022).

19. B. J. Hilton, A. Husch, B. Schaffran, T. Lin, E. R. Burnside, S. Dupraz, M. Schelski, J. Kim, J. A. Müller, S. Schoch, C. Imig, N. Brose, F. Bradke, An active vesicle priming machinery suppresses axon regeneration upon adult CNS injury. Neuron. 110, 51-69.e7 (2022).

20. K. M. Berendzen, D. S. Manoli, Rethinking the Architecture of Attachment: New Insights into the Role for Oxytocin Signaling. Affect Sci. 3, 734–748 (2022).

21. C. M. Bacmeister, R. Huang, L. A. Osso, M. A. Thornton, L. Conant, A. R. Chavez, A. Poleg-Polsky, E. G. Hughes, Motor learning drives dynamic patterns of intermittent myelination on learning-activated axons. Nat Neurosci. 25, 1300–1313 (2022).

22. E. G. Hughes, J. L. Orthmann-Murphy, A. J. Langseth, D. E. Bergles, Myelin remodeling through experience-dependent oligodendrogenesis in the adult somatosensory cortex. Nat Neurosci. 21, 696–706 (2018).

23. M. A. Thornton, E. G. Hughes, Neuron-oligodendroglia interactions: Activity-dependent regulation of cellular signaling. Neurosci Lett. 727, 134916 (2020).

24. C. M. Durand, C. Betancur, T. M. Boeckers, J. Bockmann, P. Chaste, F. Fauchereau, G. Nygren, M. Rastam, I. C. Gillberg, H. Anckarsäter, E. Sponheim, H. Goubran-Botros, R. Delorme, N. Chabane, M.-C. Mouren-Simeoni, P. de Mas, E. Bieth, B. Rogé, D. Héron, L. Burglen, C. Gillberg, M. Leboyer, T. Bourgeron, Mutations in the gene encoding the synaptic scaffolding protein SHANK3 are associated with autism spectrum disorders. Nat Genet. 39, 25–27 (2007).